# IMPROVE VISION LANGUAGE MODEL CHAIN-OF-THOUGHT REASONING

## ABSTRACT

Chain-of-thought (CoT) reasoning in vision language models (VLMs) is crucial for improving interpretability and trustworthiness. However, current training recipes lack robust CoT reasoning data, relying on datasets dominated by short annotations with minimal rationales. In this work, we show that training VLM on short answers does not generalize well to reasoning tasks that require more detailed responses. To address this, we propose a two-fold approach. First, we distill rationales from GPT-4o model to enrich the training data and fine-tune VLMs, boosting their CoT performance. Second, we apply reinforcement learning to further calibrate reasoning quality. Specifically, we construct positive (correct) and negative (incorrect) pairs of model-generated reasoning chains, by comparing their predictions with annotated short answers. Using this pairwise data, we apply the Direct Preference Optimization algorithm to refine the model's reasoning abilities. Our experiments demonstrate significant improvements in CoT reasoning on benchmark datasets and better generalization to direct answer prediction as well. This work emphasizes the importance of incorporating detailed rationales in training and leveraging reinforcement learning to strengthen the reasoning capabilities of VLMs.

## 1 INTRODUCTION

Chain-of-thought (CoT) reasoning is essential for improving the interpretability and trustworthiness of VLMs (Li et al., 2024; Liu et al., 2024; Chen et al., 2023; Liu et al., 2023b;a; Bai et al., 2023). As VLMs are increasingly applied to more difficult tasks, the ability to reason through complex problems becomes essential. However, current training approaches for VLMs often rely on datasets dominated by short answers with limited rationales, which may restrict the models' ability to generalize to tasks with comprehensive reasoning. In this work, we aim to address these limitations by providing distilled CoT data, introducing supervised finetuning (SFT) and reinforcement learning (RL) strategies to improve VLM reasoning performance.

An example in fig. 1 asks for the number of food items in a bar graph. When answering this question, a human would typically enumerate the bars and then calculate the total. However, writing out this enumeration process is far more cumbersome than simply providing the short answer of "14." Consequently, the annotated training data is predominantly composed of short answers, with minimal rationale provided. This raises a critical research question: *Does training on direct prediction implicitly teach the model to perform chain-of-thought reasoning to derive correct answers?* Our findings indicate that after training on 26k direct predictions from ChartQA, the accuracy of direct predictions increased by 2.9 (70.2 to 73.1), while CoT prediction accuracy improved by only 0.6 points (71.2 to 71.8), with CoT under-performing direct prediction as a result. This suggests that current training approaches have limited effectiveness in enhancing CoT reasoning.

We hypothesize that developing CoT reasoning capabilities requires explicit training on data that includes detailed reasoning steps. To address the scarcity of high quality CoT reasoning data, we propose leveraging datasets with short ground truth annotations and employing the GPT-4o model to generate reasoning paths that lead to the correct answer. Our approach encompasses a diverse range of tasks, utilizing 9 datasets that demand different reasoning skills, including common world knowledge (A-OKVQA), chart interpretation (ChartQA), document information localization (DocVQA, InfoVQA), real-world text extraction (TextVQA), scientific reasoning (AI2D, SQA), and

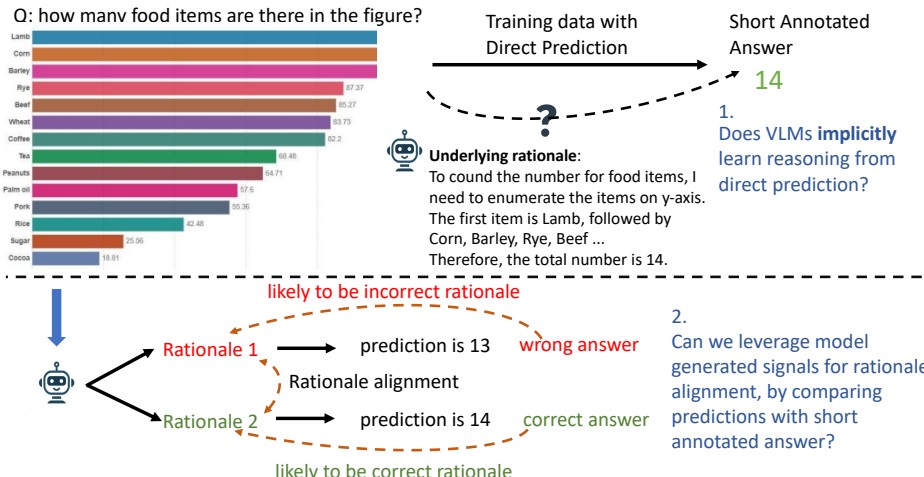

Figure 1: The upper figure questions whether training exclusively on direct-answer datasets can effectively teach CoT prediction. In the lower figure, generating CoT for prediction provides the additional benefit of reasoning alignment, allowing the model to improve by leveraging self-generated data.

mathematical reasoning (MathVision, G-LLaVA). We distilled a total of 193k CoT examples for supervised fine-tuning (SFT) and the model, LLAVA-REASONER-SFT, demonstrates significant improvements in VLM chain-of-thought reasoning performance.

In the lower part of fig. 1, we propose further refining SFT model reasoning through model-generated signals (Sun et al., 2024; Setlur et al., 2024). Specifically, the model generates multiple CoT steps to derive final predictions, which are then compared to the provided short annotation. Rationales that lead to correct predictions are more likely to be accurate, and vice versa. By optimizing positive (correct) and negative (incorrect) pairs of rationales with Direct Preference Optimization (DPO), we align the VLM toward more accurate reasoning process. The aligned model, LLAVA-REASONER-DPO, shows improved performance across all three domains as well as better out-of-domain generalization. Additionally, we demonstrate that DPO model can serve as a verifier to assign appropriate rewards, facilitating effective credit assignment (Rafailov et al., 2024; Lu et al., 2024).

Our contributions can be summarized as follows: (A) We release a comprehensive CoT dataset SHAREGPT-4O-REASONING containing 193k examples, covering various VQA tasks. (B) We demonstrate the effectiveness of SFT in improving CoT reasoning using this dataset. (C) We show that reinforcement learning with DPO can further improve model reasoning using model-generated signals, without requiring additional human-labeled data.

## 2 RELATED WORK

**VLM Reasoning** Previous work has evaluated the reasoning capabilities of vision-language models (VLMs) across various domains, including mathematics (Lu et al., 2023; Wang et al., 2024), college-level questions (Yue et al., 2024), and science (Kembhavi et al., 2016; Lu et al., 2022). Training free methods introduce scene decomposition (Mitra et al., 2024) or additional coarse and fine-grained localization (Luan et al., 2024) to improve visual reasoning. Studies such as Zhang et al. (2024c;a); Gao et al. (2023) focus on training VLMs to generate step-by-step solutions for math problems or chart-based calculations. Shao et al. (2024) improves VLMs CoT by highlighting bounding box essential for answering the related questions.

**VLM/LLM Alignment** VLM alignment has utilized preference modeling techniques, such as Direct Preference Optimization (DPO)(Ouali et al., 2024; Deng et al., 2024; Yu et al., 2024; Li et al., 2023; Gunjal et al., 2023; Sun et al., 2023), and Proximal Policy Optimization (PPO)(Sun et al., 2023), to improve factual accuracy and reduce hallucination. To improve reasoning capabilties in LLM, Sun et al. (2024); Setlur et al. (2024); Lu et al. (2024); Pang et al. (2024); Xie et al. (2024) use iterative or step DPO to improve math CoT reasoning capabilities.

**(A) Rationale Distillation Given Short Annotation**

Generate a rationale for the answer

GPT-4o

How to get to Tech Camps?
Answer: turn right

Rationale:
In the image, we see a directional sign at the bottom that points towards "Tech Camps." The sign is green with a white border and a white arrow, indicating the direction in which one should navigate to reach the camps. The arrow on the sign is clearly pointing to the right. This suggests that to get to Tech Camps, one must follow the direction indicated by the right-pointing arrow on the sign.

**(B) SFT with CoT Data**

Input: {image + question}
Output: {rationale + answer}

SFT

Sample

**Rollout Generation (Rationale + Prediction)**

Rationale

Based on the image you provided, the sign for "iD Tech Camps" is ...

This image shows a sign of ...

Predicted answer

Answer: B

Answer: A

**(C) Preference Data Construction for DPO**
Use the correctness of predicted answer as feedback.
Positive: reasoning that leads to correct answer.
Negative: reasoning that leads to wrong answer

Figure 2: Workflow diagram showing: a) the use of GPT-4o to generate rationale given short annotations; b) SFT of open-source VLM for CoT reasoning; c) Build preference dataset for reinforcement learning with DPO to enhance reasoning.

# 3 METHOD

As shown in fig. 2, our pipeline consists of three stages: (A) CoT data distillation from GPT-4o (section 3.1), (B) SFT with CoT (and direct) data to enable VLM CoT reasoning, and (C) RL for further enhancement of CoT reasoning. The RL stage involves generating positive (correct) and negative (incorrect) reasoning data pairs sampled from SFT, as detailed in section 3.3.

## 3.1 REASONING DATA DISTILLATION

To mitigate the limited availability of high-quality CoT data, we leverage VQA datasets with short annotations and augment them with rationales generated by the GPT-4o model. We collect 193k visual CoT instances to create the SHAREGPT-4O-REASONING dataset, which we plan to release for public use. We focus on the following reasoning types as demonstrated in fig. 4:

**Real-World Knowledge** includes the A-OKVQA dataset (Schwenk et al., 2022), which covers a broad range of commonsense reasoning and real-world knowledge for answering questions.

**Chart Understanding** includes the ChartQA dataset (Zhang et al., 2024a), which involves tasks like item comparison, counting, and numerical computation.

**Document Understanding/Real-World Text** includes DocVQA (Mathew et al., 2021), InfoVQA (Mathew et al., 2022), and TextVQA (Singh et al., 2019), focusing on information localization and extraction in industrial documents and real-world image comprehension.

Table 1: Data statistics of CoT distilled on different dataset

| Dataset | Dataset Size |
|---------|--------------|
| A-OKVQA | 16.9k |
| ChartQA | 26.0k |
| SQA | 6.1k |
| AI2D | 11.9k |
| InfoVQA | 22.4k |
| DocVQA | 37.3k |
| TextVQA | 29.7k |
| MathVision | 11.0k |
| G-LLaVA | 30.3k |
| **Total** | **193k** |

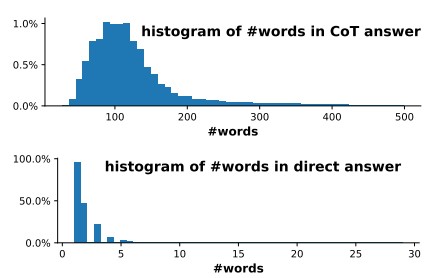

Figure 3: The distribution of word counts for CoT and direct answer.

**Math and Science** includes MathVision (Wang et al., 2024), G-LLaVA (Gao et al., 2023), SQA (Lu et al., 2022), and AI2D (Kembhavi et al., 2016), focusing on scientific knowledge and mathematical reasoning.

After distillation, we filtered out examples whose answer predicted by GPT-4o is different from ground truth. The data statistics are presented in table 1, and a comparison of answer lengths is shown in fig. 3, highlighting that CoT responses peak around 100 tokens, while direct answers are typically under 5 tokens. The exact distillation prompt is provided in appendix A.

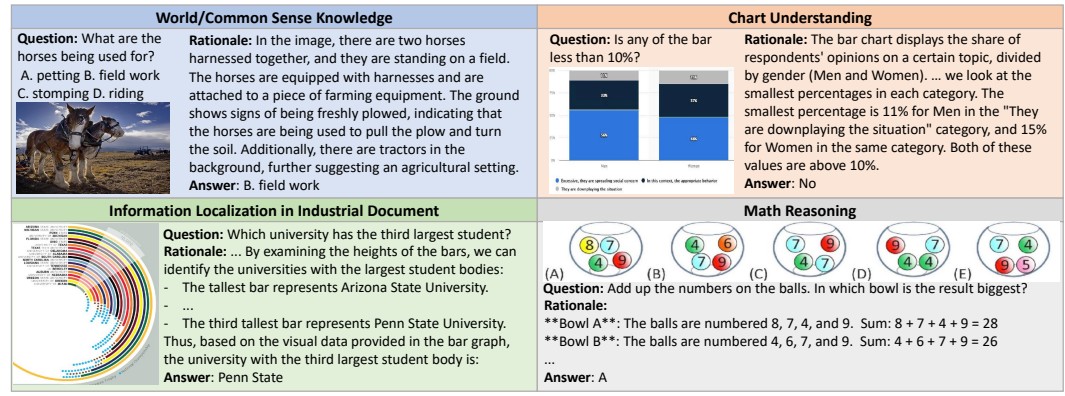

Figure 4: Distillation of examples from various VLM task domains, highlighting the specific reasoning capabilities required.

## 3.2 SUPERVISED FINE-TUNING FOR CHAIN-OF-THOUGHT PREDICTION

We choose LLaMA3-LLaVA-NeXT-8B as our base architecture, whose weight is initialized with the Open-LLaVA-NeXT weights[1]. To ensure the model handles both direct and chain-of-thought (CoT) predictions, we implement two types of prompts during training.

**Direct Prediction:** For direct prediction tasks, we use the prompt "Answer the question with a short answer" for short-answer questions, and "Answer with the option's letter from the given choices directly" for multiple-choice questions.

**CoT Prediction:** For CoT prediction tasks, we use the prompt "Generate a reason first and then output a letter answer" for multiple-choice questions, and "Generate a reason first and then output a short answer" for short-answer questions. In the model's response, the rationale is followed by the answer, which is formatted as "### Answer: " to enable answer extraction during evaluation.

## 3.3 REINFORCEMENT LEARNING FOR ENHANCED REASONING

To further improve the quality of reasoning chains, we apply RL using the DPO algorithm to better align the model's reasoning process toward more accurate predictions. The DPO algorithm requires both positive and negative responses. To generate these, we use the SFT model as the policy model (i.e., generator), producing 32 candidate predictions per question (temperature 1.0 for short answer and 1.2 for multiple-choice questions). Each prediction is compared with the ground truth to determine its correctness (fig. 2). Following the approach in Dubey et al. (2024), we select instances with an accuracy between 0.25 and 0.85. From these, we randomly pair positive and negative responses, creating up to three pairs per question.

Formally, the dataset is denoted as $\mathcal{D}_{DPO} = \{(\mathcal{V}, x, y_w, y_l)\}$, where $\mathcal{V}$ is the image, $x$ is the question, $y_w$ and $y_l$ are the positive and negative responses. The DPO objective is defined as below:

$$\mathcal{L}_{\text{DPO}}\left(\pi_\theta; \pi_{\text{ref}}\right) = -\mathbb{E}_{(\mathcal{V}, x, y_w, y_l) \sim \mathcal{D}_{DPO}} \left[ \log \sigma \left( \beta \log \frac{\pi_\theta\left(y_w \mid x, \mathcal{V}\right)}{\pi_{\text{ref}}\left(y_w \mid x, \mathcal{V}\right)} - \beta \log \frac{\pi_\theta\left(y_l \mid x, \mathcal{V}\right)}{\pi_{\text{ref}}\left(y_l \mid x, \mathcal{V}\right)} \right) \right],$$

where $\pi_\theta$ is the policy model to be optimized and $\pi_{\text{ref}}$ is the base reference model, both models are initialized with SFT weights. $\sigma$ is the logistic function and $\beta$ is set to 0.1.

---

[1]https://github.com/xiaoachen98/Open-LLaVA-NeXT

Table 2: SFT experiments with data composition in fig. 5: ① format alignment only, ② direct responses only, ③ CoT responses only and ④ both direct and CoT responses. Inference is performed using both direct and CoT templates. The best CoT prediction result is highlighted in orange, while the best direct prediction result is marked in blue. The results demonstrate that combining CoT and direct responses during training leads to the best performance across both types of prompts. Refer to section 4 for detailed analysis.

| Methods | Prompting | A-OK | ChartQA | DocVQA | InfoVQA | TextVQA | AI2D | SQA | MathVista | Avg |
|---------|-----------|------|---------|--------|---------|---------|------|-----|-----------|-----|
| LLaVA-Next | direct | 85.8 | 70.2 | 75.7 | 37.7 | 68.2 | 71.5 | 75.4 | 39.3 | 65.5 |
| + Format ① | CoT | 84.3 | 71.2 | 67 | 34.9 | 62.2 | 67.4 | 74.4 | 40.3 | 62.7 |
| LLaVA-Next | direct | 86.4 | 73.7 | 78 | 45.4 | 71.9 | 78.8 | 91.5 | 43.2 | 71.1 |
| + Direct ② | CoT | 85.7 | 71.8 | 68.8 | 38.6 | 63.6 | 72.5 | 85.4 | 38.6 | 65.6 |
| LLaVA-Next | direct | 84.9 | 71.8 | 81.2 | 45.7 | 72.1 | 75.3 | 85 | 41.9 | 69.7 |
| + Cot ③ | CoT | 85.1 | 82.2 | 81.2 | 49.7 | 69.9 | 77 | 91.3 | 49.2 | 73.2 |
| LLaVA-Reasoner | direct | 85.4 | 76.1 | 82.9 | 50.6 | 73.1 | 79.4 | 90.4 | 44.3 | 72.8 |
| -SFT ④ | CoT | 86.2 | 83.0 | 81.8 | 51.6 | 71.1 | 78.5 | 92.7 | 50.6 | 74.4 |

# 4 SFT EXPERIMENTS FOR CHAIN-OF-THOUGHT LEARNING

In this section, we explore how SFT can enhance VLM reasoning by addressing two key research questions: (1) *Can CoT reasoning be implicitly learned from short responses?* and (2) *How effectively can CoT be learned from GPT-4o distilled data?* Additionally, we analyze the composition of CoT data across various reasoning capabilities and compare the performance of SOTA models with GPT-4o.

## 4.1 TRAINING SETTING

As shown in the upper part of fig. 5, we present the data composition for SFT. The training data includes CoT distillation (193k instances) from table 1 and corresponding short answers (193k). Additionally, for CoT data, we incorporate 16k visual math examples from G-LLaVA. To main-

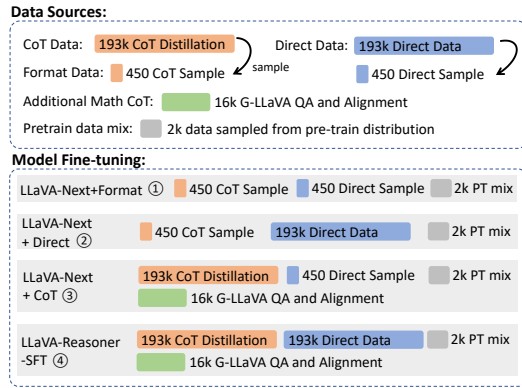

Figure 5: The upper section displays the data sources used for the SFT experiments, while the lower section illustrates the data composition for model training.

tain general instruction-following capability as the base model, we include 2k randomly sampled instruction data from LLaVA pretraining Liu et al. (2024). To ensure the SFT models can handle both direct and CoT prompts during inference, we sample a small set of format-aligned data—50 examples from each of the 9 datasets—resulting in 450 instances.

In the lower part of fig. 5, we outline the data composition for model training. Specifically, LLAVA-NEXT-FORMAT (fig. 5 ①) serves as the baseline model, trained exclusively on format-aligned data to enforce the desired output format without learning any task-specific reasoning skills. In contrast, models in fig. 5 ② and ③ incorporate either direct or CoT datasets, enabling the model to be expert in one type of skill as well as following the both direct and CoT prompt styles. Finally, LLAVA-REASONER-SFT (fig. 5 ④) represents the SFT model trained on both CoT and direct data, making it to be expert in both types of reasoning.

We use the LLaMA3-LLaVA-NeXT-8B architecture, initializing the weights with Open-LLaVA-NeXT. All Supervised Fine-Tuning (SFT) experiments are trained for 1 epoch with a learning rate of 5e-6 and a batch size of 32. The experiments are conducted on 8 H100 GPUs.

## 4.2 EVALUATION SETTING

We evaluate our method using a range of benchmark datasets, including A-OKVQA (Schwenk et al., 2022), ChartQA (Masry et al., 2022), DocVQA (Mathew et al., 2021), InfoVQA Mathew et al. (2022), TextVQA (Mathew et al., 2021), AI2D (Kembhavi et al., 2016), ScienceQA (Lu et al., 2022), and MathVista (Lu et al., 2023). We also conduct more evaluation on general datasets OCRBench (Liu

et al., 2023c), MMStar (Chen et al., 2024a), and MMMU (Yue et al., 2024) in later sections. The evaluation for A-OKVQA was implemented by us, while for the other datasets, we follow the evaluation protocols outlined in VLMEval (Duan et al., 2024).

For CoT evaluation, answers are extracted after the pattern "###Answer: " before sent to evaluation. More comparison with LLaMA3-LLaVA-NeXT-8B model is shown appendix C and evaluation on GPT-4o is shown in appendix B.

### 4.3 CAN REASONING BE IMPLICITLY LEARNT FROM DIRECT PREDICTION?

Table 2 presents the performance of the models introduced in fig. 5. Since LLAVA-NEXT-8B training data contains very few CoT reasoning examples, CoT performance of ① lags behind direct prediction across most tasks. The only improvement is observed in ChartQA and MathVista with a modest gain of +1.0 in CoT performance, showing CoT is helpful for calculation related tasks.

When comparing model trained on direct only data (②) to that trained on format-aligned data (①), we observe an average gain of +5.6 in direct prediction accuracy ($65.5 \rightarrow 71.1$) and a +2.9 improvement in CoT performance ($62.7 \rightarrow 65.6$). Surprisingly, closer inspection of CoT performance in calculation-involved tasks, such as ChartQA and MathVista, reveals only marginal gains (+0.6 for ChartQA CoT) or even a performance drop (-1.7 on MathVista), which contrasts with the improvements seen on the two tasks in ①. On text-rich tasks, positive gains (>1) are observed, with the most improvement seen in InfoVQA (+3.7). Significant gains are also evident in science-related tasks like AI2D (+5.1) and SQA (+11.0). Despite these improvements, CoT performance still trails behind direct prediction overall (CoT: 65.6 vs. direct: 71.1). This result suggests that training on direct only prediction may not effectively help with CoT prediction.

### 4.4 HOW EFFECTIVE IS CoT REASONING DATA?

When comparing the model trained on CoT-only data (③) with the one trained on format-aligned data (①), we observe improvements in both direct and CoT predictions. Direct prediction performance increases by an average of +4.2 ($65.5 \rightarrow 69.7$), while CoT prediction improves significantly by +10.5 ($62.7 \rightarrow 73.2$). Notably, the CoT performance of the model ③ surpasses its direct prediction (73.2 CoT vs. 69.7 direct). Significant gains are observed in calculation-intensive tasks like ChartQA and MathVista, with increases of +11.0 and +8.9 in CoT performance, respectively. Interestingly, for text-rich tasks such as DocVQA, InfoVQA, and TextVQA, the direct performance of model ③ (trained on CoT-only data) outperforms that of model ② (trained on direct-only data). This suggests that even for text-heavy tasks, reasoning processes, such as localizing information in documents or recognizing text in real-world scenarios, may benefit from CoT training. The skills learned from CoT training appear to generalize to direct prediction as well.

When both CoT and direct data are combined (④), performance is further enhanced for both prediction types, with an average gain of +7.3 in direct prediction ($65.5 \rightarrow 72.8$) and +11.7 in CoT prediction ($62.7 \rightarrow 74.4$). This demonstrates that combining direct and CoT data yields the best overall performance. Interestingly, in model ④, for 3 out of 8 datasets (TextVQA, DocVQA, AI2D), direct prediction outperforms CoT prediction. We hypothesize that these tasks involve a significant proportion of concise fact extraction, where generating long-form CoT responses may not provide additional benefits or even hurts. Further validation of this hypothesis will be explored in future work.

Table 3: Effect of data composition on math reasoning. MV: MathVision, GL: G-LLaVA, MI: MathInstruct, MP: Math-Plus.

| Data Config | MathVista (direct/CoT) |
|---|---|
| format only ① | 39.3/40.3 |
| MV | 41.0/43.4 |
| MV+GL | 43.2/44.9 |
| MV+GL+MP50k | 42.3/45.6 |
| MV+GL+MP100k | 43.0/44.9 |
| MV+GL+MI50k | 43.1/45.0 |
| MV+GL+MI100k | 43.7/46.3 |
| MV+GL+AI2D | 44.1/46.4 |
| MV+GL+SQA | 43.1/47.3 |
| MV+GL+ChartQA | 43.2/50.4 |

Table 4: Effect of data composition on science related tasks.

| Data Config | AI2D | SQA |
|---|---|---|
| format only ① | 67.4 | 74.4 |
| AI2D | 76.3 | 76.6 |
| SQA | 66.9 | 90.4 |
| AI2D +SQA | 76.7 | 91.2 |
| AI2D +SQA +ChartQA | 77.4 | 91.4 |

## 4.5 ABLATION TESTS ON DATA COMPOSITION

**Data Composition for Math.** In table 3, we examine the effectiveness of data composition on MathVista performance. We first include two visual math datasets: MathVision (MV) and G-LLaVA (GL). Including MV improves CoT performance by +3.1 over format only baseline (fig. 5 ①), while adding GL yields an additional gain of +1.5. Building on MV+GL, we incorporate several datasets that are potentially relevant to the task, including two math text-only datasets: MathPlus (MP) and MathInstruct (MI), two science datasets: SQA and AI2D, and ChartQA. Notably, ChartQA significantly boosts CoT performance (+5.5), while AI2D and SQA provide positive gains of +0.6 and +1.5, respectively. However, adding the math text datasets results in minimal improvement. Comparing inclusion of 100k MP vs 50k MP, more text data does not necessarily lead to better results. Therefore, we decided not to include them in training LLAVA-REASONER-SFT.

**Data Composition for Science Tasks with CoT Prediction.** In table 4, we evaluate the impact of data composition on science datasets, including AI2D and SQA. Our results show that combining SQA and AI2D provides additional gains on both datasets, indicating that they are mutually beneficial. Furthermore, adding ChartQA contributes positively to both datasets, with a notable improvement of +0.7 for AI2D.

## 4.6 COMPARING WITH SOTA MODEL AND GPT-4O

In table 5, we compare the performance of GPT-4o and a recent state-of-the-art model, Cambrian Tong et al. (2024). For GPT-4o, we include both direct and CoT predictions, following the prompt optimization steps outlined in Borchmann (2024), with the prompts detailed in appendix B. For Cambrian, we report the numbers from Tong et al. (2024) and replicated the results using the official checkpoint on MMStar, InfoVQA, and A-OKVQA. Specifically for Cambrian, CoT predictions were used for the MathVista dataset, while direct predictions were applied for the remaining datasets.

When compared to open-source models, GPT-4o outperforms on nearly all benchmark datasets, with the exception of SQA. Notably, significant improvements from CoT predictions are observed on tasks involving calculation or complex reasoning, such as ChartQA, MathVista, MMMU, and MMStar.

Table 5: Performance Comparison of GPT-4o, Cambrian-7b, and our SFT Model. For Cambrian, * indicates our replicated results, while others are adapted from Tong et al. (2024), † indicate CoT prompt used for evaluation. 'Our-SFT' refers to LLAVA-REASONER-SFT.

| Dataset | GPT-4o direct/cot | Cambrian official | Our-SFT direct/cot |
|---------|-------------------|-------------------|--------------------|
| A-OK | 89.6/90.1 | 83.1* | 85.4/86.2 |
| ChartQA | 79.6/84.7 | 73.3 | 76.1/83.0 |
| DocVQA | 90.3/90.8 | 77.8 | 82.9/81.8 |
| InfoVQA | 72.4/72.8 | 45.7* | 50.6/51.6 |
| TextVQA | 78.1/75.4 | 71.7 | 73.1/71.1 |
| AI2D | 80.7/81.5 | 73.0 | 79.4/78.5 |
| SQA | 85.9/87.2 | 80.4 | 90.4/92.7 |
| MathVista | 54.8/63.4 | 49.0† | 44.3/50.6 |
| OCRBench | 80.2/79.2 | 62.4 | 61.6/62.0 |
| MMStar | 55.1/64.7 | 50.3* | 51.6/54.0 |
| MMMU | 57.8/63.6 | 42.7 | 41.6/40.0 |
| Avg (of best) | 77.9 | 64.5 | 68.8 |

Cambrian-7B is trained on a dataset of 7 million open-source instruction-following examples. In contrast, our model, fine-tuned on fewer than 400k instruction examples, outperforms Cambrian-7B on most benchmark datasets, underscoring the effectiveness of incorporating CoT data. While we recognize the challenge of comparing against other models, such as One-Vision (Li et al., 2024), MiniCPM-V Yao et al. (2024), X-Composer Zhang et al. (2024b), and InternVL Chen et al. (2024b), due to differences in model architecture, training datasets, and evaluation pipelines, our primary focus is on studying the effectiveness of CoT learning rather than competing for state-of-the-art performance on visual-language tasks.

## 5 RL EXPERIMENTS FOR ENHANCED CHAIN-OF-THOUGHT REASONING

In this section, we demonstrate the effectiveness of RL in further enhancing CoT reasoning. We employ the DPO algorithm, which is directly optimized using positive and negative pairs. By leveraging short-answer feedback (section 3.3), we construct preference pairs across three domains:

Table 6: DPO experiment with LLaVA-REASONER-SFT as the base policy model. We compare two DPO datasets: ⑤ RLAIF-V Yu et al. (2024) and ⑥ our preference dataset comprising A-OKVQA, ChartQA, and math. The best CoT prediction is highlighted in orange. Our DPO dataset shows the better improvements in chain-of-thought reasoning.

| Methods | Prompting | A-OK | ChartQA | DocVQA | InfoVQA | TextVQA | AI2D | SQA | MathVista | Avg |
|---|---|---|---|---|---|---|---|---|---|---|
| LLaVA-Reasoner | direct | 85.4 | 76.1 | 82.9 | 50.6 | 73.1 | 79.4 | 90.4 | 44.3 | 72.8 |
| -SFT ④ | CoT | 86.2 | 83.0 | 81.8 | 51.6 | 71.1 | 78.5 | 92.7 | 50.6 | 74.4 |
| LLaVA-Reasoner | direct | 85.6 | 76.1 | 83.1 | 50.7 | 73.3 | 79.6 | 91.1 | 44.1 | 73.0 |
| -RLAIF ⑤ | CoT | 86.7 | 83.0 | 82.4 | 50.8 | 71.4 | 79.1 | 92.9 | 50.8 | 74.6 |
| LLaVA-Reasoner | direct | 85.4 | 76.4 | 83.1 | 51.2 | 73.3 | 79.4 | 90.8 | 44.2 | 73.0 |
| -DPO-ours ⑥ | CoT | 87.0 | 84.2 | 82.7 | 52.7 | 71.5 | 79.5 | 92.6 | 52.1 | 75.3 |

A-OKVQA (real-world knowledge reasoning), ChartQA (chart interpretation), and math (MathVision and G-LLaVA). Although additional DPO data from other datasets could be incorporated, data scaling and balancing will be addressed in future work.

For the DPO dataset, we include 24.5k examples from ChartQA, 18.3k from A-OKVQA, and 22.0k from math domain, totaling 64.8k preference data pairs. We train LLaVA-REASONER-SFT on this dataset using a learning rate of 5e-7, a batch size of 32, and for 1 epoch. We found an additional trick to truncate the responses up to 90 tokens to be helpful for DPO training. To compare the effectiveness of different DPO datasets, we include RLAIF-V Yu et al. (2024), which contains 80k DPO pairs representing the state-of-the-art dataset for aligning VLMs for reducing hallucinations.

## 5.1 CAN DPO CALIBRATE REASONING?

In table 6, we present the results of the DPO model optimized on top of LLaVA-REASONER-SFT (④). Model ⑤ uses the SOTA RLAIF-V Yu et al. (2024) data, while model ⑥ uses our dataset. We observe that Model ⑤ shows a slight improvement in both direct prediction (+0.2) and CoT prediction (+0.2), whereas model ⑥ demonstrates a greater improvement in CoT prediction (+1.1) with equal gains on direct prediction. Interestingly, though only 3 out of 8 datasets are selected to construct DPO pairs, gains are observed across 7 out of 8 datasets except for SQA with a slight decrease (92.9 → 92.6). These results suggest that DPO dataset constructed from model-generated rationales can effectively enhance reasoning accuracy and show generalization across tasks.

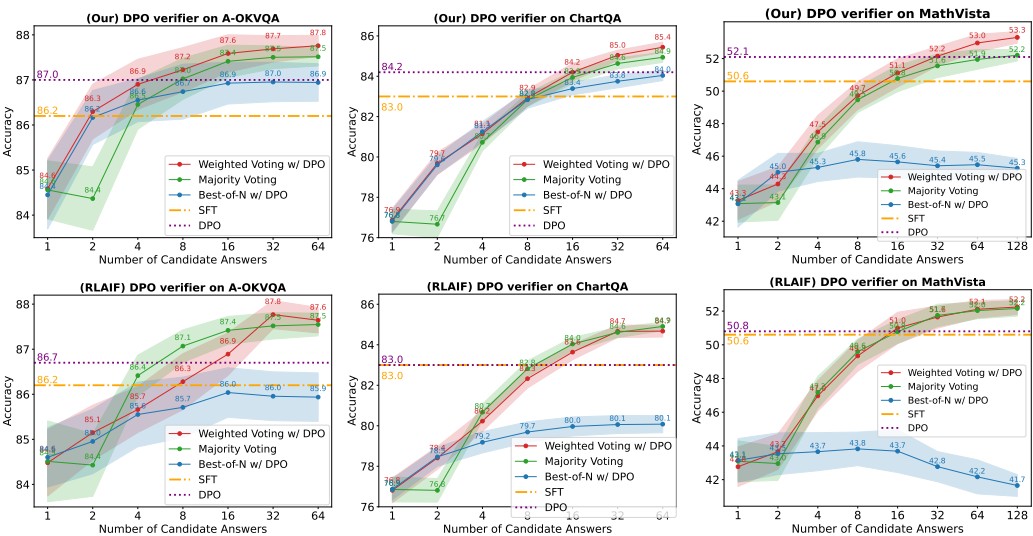

Figure 6: The figures illustrate the performance of the DPO model as a verifier on ChartQA, A-OKVQA, and MathVista. Compared to the model trained with RLAIF-V, the model trained on our reasoning data pairs consistently shows improvement in both best-of-N selection and weighted voting.

## 5.2 DPO as Verifier for CoT Reasoning Re-ranking

In fig. 6, we present the re-ranking results using the DPO model as a verifier, following the approach of Zhang et al. (2024d); Hosseini et al. (2024); Lu et al. (2024). The DPO reward score is calculated as $\log \frac{\pi_{\text{dpo}}(y|x,\mathcal{V})}{\pi_{\text{sft}}(y|x,\mathcal{V})}$, where $\mathcal{V}$ represents the image, $x$ the question, and $y$ the candidate answer. We explore two re-ranking strategies: Best-of-N and Weighted Voting. A Majority Voting (or self-consistency) baseline is also included for comparison.

When trained with RLAIF-V data (⑤), the DPO model demonstrates improvements as both a generator and verifier on A-OKVQA, likely due to the dataset's alignment with real-world images, which matches the nature of A-OKVQA. Interestingly, while model ⑤ does not show improvements as a genera-

Table 7: More DPO results on general evaluation benchmark datasets.

| Methods | OCRBench | MMStar | MMMU | Avg |
|---|---|---|---|---|
| SFT ④ | 62.0 | 54.0 | 40.1 | 52.0 |
| SFT+RLAIF ⑤ | **63.7** | 53.5 | 42.3 | 53.2 |
| SFT+DPO-ours ⑥ | **63.7** | **54.1** | **42.6** | **53.5** |

tor on ChartQA, it still produces positive results in best-of-N re-ranking, indicating that the learned preferences can generalize across domains. However, weighted voting does not lead to any improvements, and no significant gains are observed in re-ranking for MathVision. In contrast, when trained with reasoning data pairs, LLaVA-Reasoner-DPO (⑥) shows improvements across both re-ranking metrics, underscoring the effectiveness of DPO on reasoning data pairs.

## 5.3 Additional DPO CoT Performance on General Datasets

In table 7, we present the DPO CoT performance on OCRBench, MMStar, and MMMU. We observe that both DPO models outperform the SFT baseline, with our DPO model trained on CoT reasoning pairs showing slightly better results.

In fig. 7, we further explore the effectiveness of DPO on the MMMU dataset, which consists of challenging college-level subject questions. We provide re-ranking results for multiple-choice problems from the Dev+Val split (988/1050). First, the SFT model with self-consistency shows consistent improvements reaching 45.5 with 64 candidate votes. LLaVA-Reasoner-DPO, trained on reasoning data pairs, shows strong generalization on MMMU by excelling in both weighted voting and best-of-N voting during candidate re-ranking. While the DPO model trained on RLAIF-V (⑤) improves CoT predictions, it does not achieve gains in the re-ranking metrics, indicating limitations in distinguishing correct from incorrect reasoning on more complex data. We hypothesize that, compared to ChartQA, the reasoning questions in MMMU are more challenging and span a broader range of subjects. The RLAIF-V dataset relies primarily on COCO image domain, which may not provide sufficient coverage, leading to weaker performance in re-ranking.

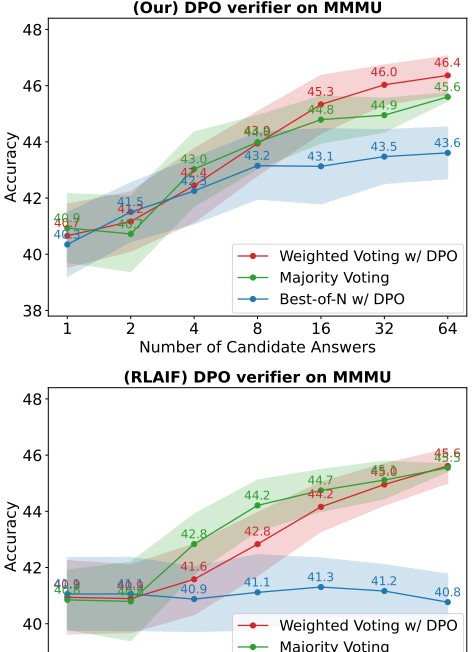

Figure 7: The performance of the DPO verifier on the MMMU dataset with 988 multiple-choice questions. We observe that DPO trained on our reasoning dataset achieves consistent improvements in re-ranking metrics, while DPO trained with RLAIF does not show significant gains.

## 5.4 DPO Credit Assignment

While the DPO model is trained on pairwise data, prior works (Rafailov et al., 2024; Lu et al., 2024)

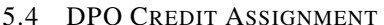

have shown that DPO policies can learn to predict *token-level rewards* from binary preference data. These experiments primarily focused on math reasoning with LLMs. In this work, we provide

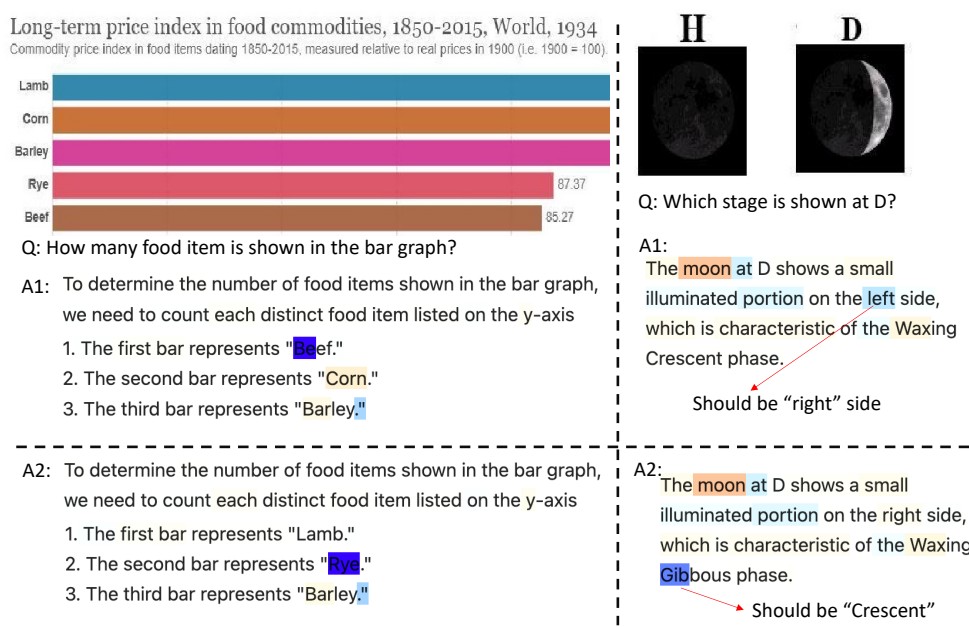

Figure 8: Credit assignment of the DPO model on a portion of the responses from the ChartQA and AI2D datasets. The DPO token-level reward is computed for each token, with the rewards normalized to have a mean of 0. Negative scores are highlighted in cool colors (blue), while positive scores are highlighted in warm colors (orange). We observe that the DPO model is particularly sensitive to the first mistakes or hallucinations introduced in the response.

examples of credit assignment learned by the VLM DPO, as shown in fig. 8. The token-level DPO reward can be expressed as $\log \frac{\pi_{\text{dpo}}(y_i|x,\mathcal{V})}{\pi_{\text{sft}}(y_i|x,\mathcal{V})}$, where $\mathcal{V}$ represents the image, $x$ the question, and $y_i$ the $i$-th token in the generated response. This reward reflects the relative confidence of the DPO model compared to the SFT model for a given token in a candidate response.

In fig. 8, negative scores are shown in cool (blue) colors, while positive scores are shown in warm (orange) colors, with rewards normalized to a mean of 0. On the left, we observe that the DPO model is particularly sensitive to errors during chart interpretation from the ChartQA dataset. For instance, when the response incorrectly lists "Lamb" as "Beef" in a chart reading task, the DPO model assigns a highly negative score to this mistake.

On the right, we present examples from the AI2D dataset. Here, a hallucination in the response, such as incorrectly stating that the left side of the moon is illuminated (the correct answer is the right side), receives a low score. Additionally, when external knowledge is required to correctly identify the moon's phase as "Crescent" instead of "Gibbous," the DPO model penalizes the incorrect "Gibbous" answer with a negative score. This indicates that the DPO model is more sensitive to knowledge-based errors than the SFT model, explaining its superior performance on CoT reasoning tasks in datasets such as AI2D.

## 6 CONCLUSION

In this work, we aim to improve VLM CoT reasoning. First, we collect a CoT reasoning dataset SHAREGPT-4O-REASONING across a broad range of VQA tasks. We demonstrate that fine-tuning on this dataset significantly enhances reasoning performance. Additionally, we further improve these models using reinforcement learning with direct preference optimization, which strengthens their ability to reason and generalize to direct answer prediction tasks. Our results show that these approaches effectively enhance the reasoning capabilities of VLMs, paving the way for more robust and interpretable multimodal models.

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

# CONTENT OF APPENDIX

In this paper, we aim to enhance chain-of-thought (CoT) reasoning in visual language models. In the main paper, we have discussed the CoT data distillation, supervised-finetuning (SFT) and reinforcement learning (Rl) with direct preference optimization (DPO) algorithm. In the appendix, we provide additional items that offer further insight into each aspect:

# A SHAREGPT-4O-REASONING DATA FOR VLM COT REASONING

## A.1 PROMPT FOR GPT-4O DISTILLATION

Figure A.1 and fig. A.2 illustrate the GPT-4o system (task) prompt and the GPT-4o distillation prompt. We employ the same prompt across all VQA datasets for data distillation. Specifically, the input to the prompt consists of an image, a question, and a short answer. The short answer serves as a reference for GPT-4o to generate a CoT reasoning followed by a final answer after '### Answer'. We show a few more examples in the next subsections.

```
When provided with an image, a question, and a reference answer, generate a
chain-of-thought step that helps derive your own answer.
Your rationale should include detailed visual elements in order to derive the answer.
```

Figure A.1: GPT-4o system prompt for CoT distillation.

```
# Objective #
You are provided with an image, a question and a reference answer. Your job is to
generate a rationale that logically derives the answer from the visual clues.

#############

# Question #
{question}

#############

# Reference Answer #
{answer}

#############

# Rationale Requirement #
1. Do not state an answer at the beginning. Explain the visual clues that help to
derive the answer.
2. Don't state that the reference answer is correct or consistent to your finding.
Your are writing your own solution.
3. State your own derivation at the end with new line: ### Answer: <your answer>

#############
```

Figure A.2: GPT-4o prompt for CoT distillation.

## A.2    FILTERING MISMATCHED ANNOTATIONS IN DISTILLATION

In the GPT-4o prompt shown in fig. A.2, we treat the annotation as a *reference answer* and instruct GPT-4o to generate its own solution based on that reference. In fig. A.3 and fig. A.4, we illustrate cases where the GPT-4o-generated solution differs from the annotated answer. Upon human examination, we identified errors in the annotations. For example, in fig. A.3, there are issues such as incorrect text recognition (e.g., "dentist" misidentified as "heart") and incorrect object identification (e.g., "beer" as "water"). In fig. A.4, the annotation errors involve incorrect calculations in the left figure and miscounting in the right figure.

To ensure consistency and avoid potential errors, we filtered out examples where the GPT-4o generated answer differs from the annotated answer. In SHAREGPT-4O-REASONING, we release the SFT CoT data along with the original distillation and filtered examples for reference.

| A-OKVQA Filtered Out Example #1 | A-OKVQA Filtered Out Example #2 |
|---|---|
| **Question:** What part of the body does Dr. Eng work on?
**Options**: A. heart, B. teeth, C. ears, D. lungs
**Annotated Answer**: A. heart | **Question:** What drink has the owner of this cooler stocked up on?
**Options**: A. beer, B. soda, C. water, D. wine
**Annotated Answer**: C. water |

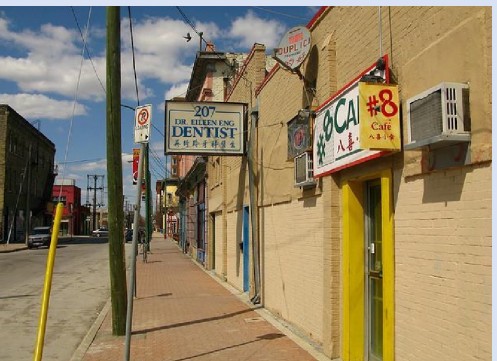

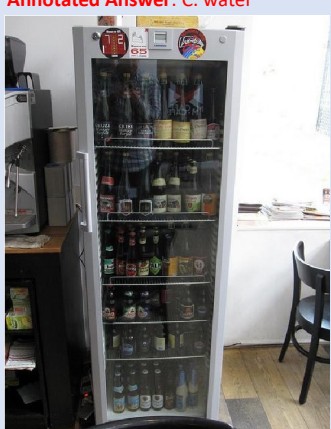

**GPT-4o Response:**
The image shows a street view with several signs on the buildings. One prominent sign reads "DR. EILEEN ENG DENTIST" along with the address "207." The word "DENTIST" is clearly visible and indicates the profession of Dr. Eileen Eng. Dentists are medical professionals who specialize in the care of teeth.

Given this information, it is clear that Dr. Eng works on teeth.

### Answer: B. teeth

**GPT-4o Response:**
The image shows a cooler with a glass door, allowing a clear view of its contents. The cooler is stocked with various bottles and cans. Upon closer inspection, the majority of the items inside the cooler appear to be bottles with labels and shapes that are typically associated with alcoholic beverages, such as beer. The bottles have different colors and designs, which are common for beer brands. There are no visible signs of soda cans, water bottles, or wine bottles. #

## Answer: A. beer

Figure A.3: An example from the A-OKVQA dataset highlights cases where the annotated answer does not match the GPT-4o-generated answer. In these cases, the GPT-4o answers are correct, while the annotations contain labeling errors. In the left figure, the sign reads "dentist" (correctly identified by GPT-4o), and the answer should relate to 'teeth,' not 'heart' as in the annotation. In the right figure, the fridge contains beer, but the annotation incorrectly labels it as 'water.' Consequently, we filter out instances where the GPT-4o-generated answer does not match the annotated answers.

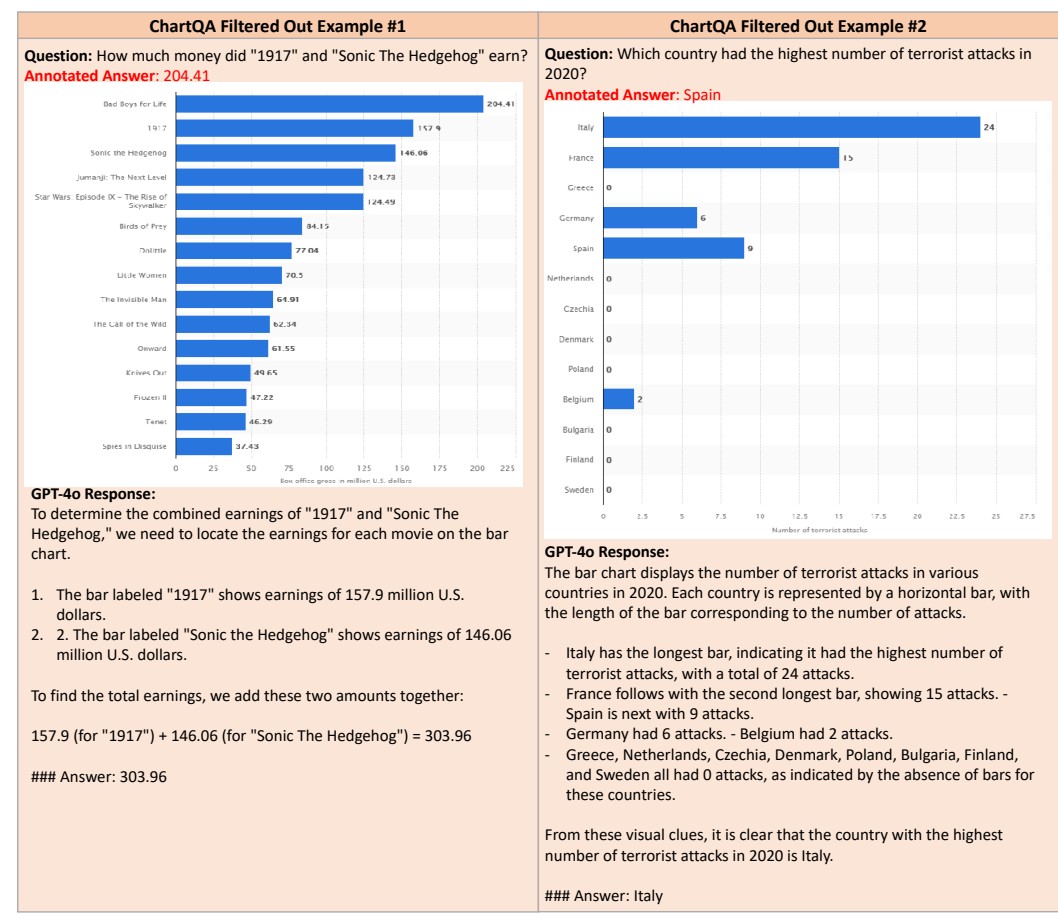

Figure A.4: Filtered examples from the ChartQA dataset are shown. In the left figure, GPT-4o correctly identifies '1917' and 'Sonic The Hedgehog' and provides the correct summation, while the annotated answer incorrectly lists '204.41', which is the value for 'Bad Boys for Life' and is unrelated to the question. In the right figure, GPT-4o accurately ranks the numbers from highest to lowest, but the annotated answer incorrectly identifies 'Spain' as having the highest value, when it should be the third largest.

# B  GPT-4O EVALUATION AND PROMPT OPTIMIZATION

In this section, we present the prompts used for GPT-4o on benchmark datasets, including both direct and Chain-of-Thought (CoT) predictions. Similar to the findings in Borchmann (2024), we observed that GPT-4o's performance is highly sensitive to prompt phrasing. We explored several sets of prompts and selected the best-performing ones for reporting results. Specifically, we try to align our results with those reported in Li et al. (2024); Tong et al. (2024), Claude 3.5 Sonnet for Vision [2], among others.

**Prompt Optimization**   We follow the process outlined in Borchmann (2024) to design effective GPT-4o prompts for the benchmark datasets. A random subset of 200 instances is selected as a development set to evaluate manually designed prompts. We manually inspect the predicted results and identify issues such as the model being overly cautious in declining answers, incorrect output formatting, or style mismatches with the ground truth labels. As an illustrative example, we detail the prompt optimization process using ChartQA, and apply similar techniques to the other datasets. Finally, we provide the prompts used for replicating our test results.

Table B.1: Prompt optimization on ChartQA for **direct** prediction evaluated with relaxed accuracy.

| # | Prompt | ChartQA (relaxed acc) |
|---|--------|-----------------------|
| 1 | `{Question}` | 2.7 |
| 2 | `{Question}`
`Answer the question directly.` | 32.3 |
| 3 | `Answer the question. Do not write a full sentence, just`
`provide a value.`
`Question: {Question}` | 56.4 |
| 4 | `Answer the question with following instruction:`
`1. Do not write a full sentence, just provide a value.`
`2. Don't include any unit, i.e. 56 instead of 56 meters`
`Question: {Question}` | 75.2 |
| 5 | `Answer the question with following instruction:`
`1. Do not write a full sentence, just provide a value.`
`2. Don't include any unit, i.e. 56 instead of 56 meters`
`3. Don't include '%' sign, i.e. 56 instead of 56%`

`Question: {Question}` | **80.3** |

We apply the prompts described in table B.1 to the development set and compare the predictions with the ground truth to optimize the prompts. Specifically, when using prompts #1 or #2, GPT-4o often generates full sentences instead of short answers. While prompt #3 produces a short answer, it often includes units or special tokens. To address this, we refined the instructions in prompt #4 by specifying that units should not be included in the final answer. This adjustment improved accuracy from 56.4 to 75.2. We also observed that the ground truth does not contain the % symbol, which could mismatch in evaluation, and we explicitly include this rule in prompt #5. Finally, we applied the tuned prompt to the test set, achieving an accuracy of 79.64 reported in table 5.

---

[2]https://www.anthropic.com/news/claude-3-5-sonnet

Table B.2: Prompt optimization on ChartQA for **CoT** prediction evaluated with relaxed accuracy.

| | **System Prompt** | **ChartQA (relaxed acc)** |
|---|---|---|
| | When provided with an image and a question, generate a rationale first and then derive an answer. Your rationale should include detailed visual elements in order to derive the answer. | |
| **#** | **Prompt** | |
| 1 | Answer the question with following instruction: 1. Generate a rationale first and then derive an answer. 2. Don't include any unit, i.e. 56 instead of 56 meters 3. Don't include '%' sign, i.e. 56 instead of 56%

Question:
{question}

# Output Format #
<rationale>
### Answer: <your answer> | **84.7** |
| 2 | Prompt #1, removing system prompt | 84.1 |

In table B.2, we first introduce output format instructions to guide GPT-4o in generating the correct CoT format, which aids in extracting the final answer. We reused the criteria from the direct prediction prompt to evaluate the results. Additionally, we found that including a system prompt typically leads to a 0.5-point increase in score across datasets, although it does not improve direct answer prediction. We hypothesize that the system prompt helps GPT-4o adhere more closely to the CoT output format. Finally, we applied the tuned prompt to the test set, achieving an accuracy of $84.72$ reported in table 5.

Following the prompt optimization steps outlined above, we provide the prompts used to replicate our GPT-4o test results in the next section.

## B.1 GPT-4O PROMPTS FOR EVALUATION

Table B.3 and table B.4 provide the optimized prompts for benchmark dataset evaluation. The tuning process does not garantee the prompt is optimal, but that roughly matches the reported value from previous papers Li et al. (2024); Tong et al. (2024), Claude 3.5 Sonnet for Vision [3], among others. We include the prompts for reference to replicate the GPT-4o results on benchmark datatsets.

Table B.3: Prompts for **direct** prediction with GPT-4o on benchmark datasets.

| Dataset | Prompt |
|---|---|
| A-OKVQA
AI2D
SQA
MMStar | Answer the question. Do not write a full sentence, just provide a letter choice.
question
{Question} |
| ChartQA | Answer the question with following instruction:
1. Do not write a full sentence, just provide a value.
2. Don't include any unit, i.e. 56 instead of 56 meters
3. Don't include '%' sign, i.e. 56 instead of 56%

Question: {Question} |
| DocVQA
TextVQA
InfoVQA
OCRBench | Answer the question. Do not write a full sentence, just provide a value.

Question: {question} |
| MathVista
MMMU | Answer the question. Do not write a full sentence, just provide a value or letter choice.
{question} |

[3]https://www.anthropic.com/news/claude-3-5-sonnet

Table B.4: Prompts for **CoT** prediction with GPT-4o on benchmark datasets.

| Dataset | CoT Prompt |
|---------|------------|
| **system prompt** | When provided with an image and a question, generate a rationale first and then derive an answer. Your rationale should include detailed visual elements in order to derive the answer. |
| A-OKVQA AI2D SQA MMStar | Answer the question with following instruction: 1. Generate a rationale first and then derive an answer. 2. For your final answer, provide a letter choice.

Question: {question}

# Output Format # \<rationale\> ### Answer: \<your answer\> |
| ChartQA | Answer the question with following instruction: 1. Generate a rationale first and then derive an answer. 2. Don't include any unit, i.e. 56 instead of 56 meters 3. Don't include '%' sign, i.e. 56 instead of 56%

Question: {question}

# Output Format # \<rationale\> ### Answer: \<your answer\> |
| DocVQA InfoVQA | # Objective # You are provided with an image, a question. Your job is to generate a rationale first and then derive an answer.

###########

# Question # {question}

###########

# Rationale Requirement # 1. Do not state an answer at the beginning. Explain descriptions of visual clue that help to derive the answer. 2. Conclude with ### Answer: \<your answer\> 3. Your final answer should be a single word or phrase. 4. If possible, copy the answer from document. Don't add or remove symbols, units, or titles.

###########

# Output Style # \<rationale\> ### Answer: \<your answer\>

########### |

Table B.4 – continued from previous page

| Dataset | **Prompt** |
|---|---|
| TextVQA | # Objective #
You are provided with an image, a question. Your job is to generate a rationale first and then derive an answer.

###########

# Question #
{question}

###########

# Rationale Requirement #
1. Do not state an answer at the beginning. Explain descriptions of visual clue that help to derive the answer.
2. Conclude with ### Answer: \<your answer>
3. Your final answer should be a single word or phrase.
4. Output your answer in lower case.

###########

# Output Style #
\<rationale>
### Answer: \<your answer>

########### |
| OCRBench | Answer the question with following instruction:
1. Generate a rationale first and then derive an answer.
2. Your answer should be a single word or phrase.

Question:
{question}

# Output Format #
\<rationale>
### Answer: \<your answer> |

Table B.4 – continued from previous page

| Dataset | Prompt |
|---------|--------|
| MathVista MMMU | ``# Objective #``
``You are provided with an image, a question. Your job is to generate a rationale that logically derives an answer from the visual clues.``

``###########``

``# Question #``
``{question}``

``###########``

``# Rationale Requirement #``
``1. Do not state an answer at the beginning. Explain step by step logic to derive the answer.``
``2. Conclude with ### Answer: <your answer>``

``###########``

``Example output style:``

``<rationale>``
``### Answer: <your answer>``

``###########`` |

# C    BASELINE EVALUATION

Table C.1: Evaluation of VLM performance on benchmark datasets with direct and CoT inference.

| Dataset | LLAVA-NEXT-8B | | LLAVA-NEXT-FORMAT | |
|---------|---------|-----|---------|-----|
| | **direct** | **CoT** | **direct** | **CoT** |
| A-OK | 85.9 | 44.5 | 85.8 | 84.3 |
| ChartQA | 68.6 | 52.8 | 70.2 | 71.2 |
| DocVQA | 78.4 | 57.1 | 75.7 | 67.0 |
| InfoVQA | 36.6 | 25.8 | 37.7 | 34.9 |
| TextVQA | 67.2 | 41.6 | 68.2 | 62.2 |
| AI2D | 73.0 | 70.0 | 71.5 | 67.4 |
| SQA | 77.4 | 75.8 | 75.4 | 74.4 |
| MathVista | 37.3 | 25.3 | 39.3 | 40.3 |
| OCRBench | 57.7 | 59.7 | 59.1 | 56.6 |
| MMStar | 47.8 | 45.7 | 44.7 | 46.7 |
| MMMU | 42.8 | 37.6 | 41.8 | 37.7 |
| Avg | 61.2 | 48.7 | 60.9 | 58.4 |

In this section, we provide evaluation details for our base model, which uses the LLAMA3-LLAVA-NEXT-8B architecture with weights initialized from OPEN-LLAVA-NEXT. We selected OPEN-LLAVA-NEXT weights because the data and training pipelines were fully available at the time of model development, allowing us to avoid reliance on the unreleased real user interactions referenced in Liu et al. (2024). The pretraining data for OPEN-LLAVA-NEXT consists of 1M image-text pairs, sourced from datasets such as ShareGPT4V, ALLaVA-Instruct-VFLAN-4V, DocVQA, SynDog-EN, ChartQA, DVQA, AI2D, and GeoQA+.

When evaluating LLAVA-NEXT-8B, we identified several issues, such as the inability to follow the CoT prompt, refusal to answer questions, and generating irrelevant reasoning. In fig. C.1, we present randomly sampled examples from LLAVA-NEXT-8B with a temperature setting of 1.0 on a ChartQA test case. These examples demonstrate the model's difficulty in adhering to the CoT prompt. In the first example, the model declines to answer the question. In the second to fourth examples, the model provides an answer first, followed by an explanation, which doesn't effectively use thought process to answer the question. In the final example, the model generates a descriptive response instead of reasoning through the question, ultimately failing to provide an answer. This illustrates the model's inconsistent handling of the prompt structure.

Table C.1 presents the evaluation results for LLAVA-NEXT-8B. For CoT predictions, we use ChatGPT to extract a letter choice or short answer from the long-form model output, using the prompts shown in fig. C.2 and fig. C.3. However, due to LLAVA-NEXT-8B 's inability to accurately follow the CoT format, its performance is significantly worse compared to direct predictions and our format-aligned model. For direct prediction, our LLAVA-NEXT-FORMAT has similar performance as that of LLAVA-NEXT-8B.

In fig. C.4, we present the same example trained with our format-aligned data for CoT using only 450 examples. The model successfully follows the CoT format by verbalizing the thought process and providing a short answer after "### Answer:". This allows us to use a rule-based extractor to retrieve answers, which also improves CoT performance, as shown in table C.1. However, the example also demonstrates that, while our data induces the CoT process, the reasoning remains incorrect. Sampling 32 examples using the format in fig. C.4 resulted in 3 correct answers, yielding an accuracy of 9.4% for that case. As a result, only slight gain is observed on ChartQA for CoT prediction vs. direct prediction.

Based on the above experiments, we report LLAVA-NEXT-8B as the baseline in the paper to avoid the difficulty in CoT evaluation and answer extraction of LLAMA3-LLAVA-NEXT-8B baseline.

---

**ChartQA LLaVA-Next-8b CoT Example**

**Question:** How many food item is shown in the bar graph?
Generate a reason first and then output a short answer.

**Sample #1:**
As the bar graph doesn't contain any numerical data or labels, it's not possible to count the exact number of food items depicted in the illustration. For a precise answer, more information or a closer examination of the specific items in the image would be necessary. In generating an accurate response, attention should be paid to the visual cues and contextual details of the scene, which might hint at the types of food items being presented.

**Sample #2:**
The bar graph displays ten different food items. These are arranged from left to right in descending order, with the first item being "Lamb." A reason for the descending order could be to represent the decreasing price index over time for that particular food item, but this is a common placement for descending bar graphs in general, so it's likely for readability of the labels.

**Sample #3:**
The bar graph illustrates nine food items, which are Lamb, Corn, Barley, Rye, Beef, Wheat, Coffee, Peanuts, Palm Oil, and Pork. Each row represents one item, and the length of the horizontal bar corresponds to the commodity price index, a measure of food prices relative to their prices in 1900.

**Sample #4:**
The bar graph shows eight food items. The reason for generating such a graph is to compare the long-term price trends of these food commodities over a century and a half, from 1850 to 2015. The graph is useful for understanding the pricing dynamics of these agricultural products, which can affect the economy, food security, and consumer affordability. The output reflects the historical price volatility and long-term increase in global food commodities, with the most significant increase in the price of grains like corn and barley, indicating their importance as staples in global food systems.

**Sample #5:**
Long-term price indices are a statistical measure of the evolution of the relative cost over time of a basket of goods. This particular bar graph is a comparison of the price indices of various food commodities over a century, representing an economic measure of the food prices in 1915 as a benchmark for relative price changes. The graph illustrates the historical cost of items in the food category, with the largest relative change indicated by the longest bars for items such as Lamb, Coffee, and Rice, and the smallest for Sugar and Cocoa.

Figure C.1: Randomly sampled examples from LLAVA-NEXT-8B with temperature=1.0 for a test case in ChartQA reveal that the model struggles to effectively follow the CoT prompt. In Sample 1, the model refuses to answer the question. In Samples 2-4, the model generates an answer first, followed by an explanation. In the final sample, the model produces a description instead of reasoning through the question, without providing an answer.

```
You are an AI assistant who will help me to match an answer with several options of a
single-choice question. You are provided with a question, several options, and an
answer, and you need to find which option is most similar to the answer. If the meaning
of all options are significantly different from the answer, output Z. Your should
output a single uppercase character in A, B, C, D (if they are valid options), and Z.
Example 1:
Question: What is the main object in image?
Options: A. teddy bear B. rabbit C. cat D. dog
Answer: a cute teddy bear
Your output: A
Example 2:
Question: What is the main object in image?
Options: A. teddy bear B. rabbit C. cat D. dog
Answer: Spider
Your output: Z
Example 3:
Question: {question}
Options: {options}
Answer: {answer}
Your output:
```

Figure C.2: ChatGPT answer extraction prompt for multiple-choices questions.

```
Your goal is to extract a short answer from a chain-of-thought prediction. You are
given a question and model prediction, the image is omitted.
You need to determine the answer from the prediction. If no answer can be derive,
output NA.

###### Example 1 ######
### Question:
How many bars are there in the chart?
### Prediction:
The result shows bar graphs ..., counting the bars, there are a total of 8 bars.
### Your output:
8

###### Example 2 ######
### Question:
Determine the date appeared in the document.
### Prediction:
The figure displays a document on financial income ... the date 2008/01/15 appears at
body of text.
### Your output:
2008/01/15

###### Your Task ######
### Question:
{question}
### Prediction:
{prediction}
### Your output:
```

Figure C.3: ChatGPT answer extraction prompt for short answer questions.

**ChartQA CoT Example of our LLaVA-NeXT-Format**

**Question:** How many food item is shown in the bar graph?
Generate a reason first and then output a short answer.

Long-term price index in food commodities, 1850-2015, World, 1934
Commodity price index in food items dating 1850-2015, measured relative to real prices in 1900 (i.e. 1900 = 100)

| Food | Value |
|---|---|
| Lamb | 103.7 |
| Corn | 103.13 |
| Barley | 102.46 |
| Rye | 87.37 |
| Beef | 85.27 |
| Wheat | 83.73 |
| Coffee | 82.2 |
| Tea | 66.48 |
| Peanuts | 64.71 |
| Palm oil | 57.6 |
| Pork | 55.36 |
| Rice | 42.48 |
| Sugar | 25.56 |
| Cocoa | 18.81 |

**Sample #1:**
The bar graph shows the long-term price index in food commodities from 1850 to 2015, measured relative to real prices in 1900. Each bar represents a different food item and the lengths of the bars are proportional to the price index values.

1.  The y-axis lists various food commodities, such as Lamb, Corn, Barley, Rye, Beef, Wheat, Coffee, Tea, Peanuts, Palm Oil, Pork, Rice, and Cocoa.
2.  Each bar along the x-axis is associated with a specific food item, and they are ordered from left to right. Counting each distinct category name along the y-axis, there are 12 different food items shown in the bar graph.

### Answer: 12

**Sample #2:**
The bar graph is a visual representation of the long-term price index of various food commodities from 1850 to 2015. Each bar is associated with a different food item, and the length of the bar represents the relative price change in terms of real prices in 1900.

By examining the horizontal axis, we can count the number of food items listed. The categories listed are Lamb, Corn, Barley, Rye, Beef, Wheat, Coffee, Peanuts, Palm Oil, Pork, Rice, Sugar, and Cocoa.

To determine the number of food items, we need to count these categories. It should be noted that even though each bar represents a specific food item, the length of the bar or the dark blue shade at the top may represent different items.

Upon counting the categories, we get the following:
1. 1 - Lamb
2. 1 - Corn
[omit]
13. 1 - Cocoa

Adding these up, we get a total of 13 food categories represented in the bar graph.

### Answer: 13

Figure C.4: Randomly sampled examples from LLAVA-NEXT-FORMAT with a temperature setting of 1.0, evaluated on the same test case in ChartQA, show that after training on 450 format-aligned data, the model is able to follow the CoT prompt by verbalizing the thought process and providing a short answer.

# D  NEARLY ZERO DATA LEARNING FOR COT REASONING

Table D.1: We study a self-taught reasoner with minimal CoT data (only 450 format-aligned examples). LLAVA-NEXT-DIRECT is used as the baseline, and our LLaVA-Next-STaR is trained with a rejection sampling method. The best CoT predictions are highlighted in orange, and the best direct predictions are highlighted in blue. Our rejection sampling method outperforms both CoT and direct prediction, with the exception of two data points.

| Methods | Prompting | A-OK | ChartQA | DocVQA | InfoVQA | TextVQA | AI2D | SQA | MathVista |
|---------|-----------|------|---------|--------|---------|---------|------|------|-----------|
| LLaVA-Next | direct | 86.4 | 73.7 | 78 | 45.4 | 71.9 | 78.8 | 91.5 | 43.2 |
| + Direct ② | CoT | 85.7 | 71.8 | 68.8 | 38.6 | 63.6 | 72.5 | 85.4 | 38.6 |
| LLaVA-Next | direct | 85.9 | 74.6 | 79.2 | 47.4 | 72.1 | 79.5 | 92.2 | 44.4 |
| -STaR | CoT | 85.9 | 77.9 | 75.8 | 44.0 | 25.1 | 76.6 | 86.8 | 42.0 |

In this section, we demonstrate how minimal CoT training data can enhance CoT reasoning capabilities. Specifically, we use only 450 CoT format-aligned examples alongside all available direct prediction data, with LLAVA-NEXT-DIRECT as the baseline. We apply rejection sampling fine-tuning (RFT) following (Sun et al., 2024; Setlur et al., 2024) to train a self-taught chain-of-thought reasoner, denoted as LLaVA-Next-STaR. From LLAVA-NEXT-DIRECT, we sample 32 CoT examples for each training instance and select those whose final predictions match the ground truth. Up to three positive examples are selected per question, resulting in a dataset of 260k RFT examples.

As shown in table D.1, RFT training improves both CoT reasoning and direct predictions overall, with the exception of two data points. Notably, TextVQA shows a significant drop in CoT performance, which we will explore further in future work. Notable (>3%) gain is observed on ChartQA, DocVQA, InfoVQA, AI2D and MathVista, and roughly 1% gain is observed on direct prediction on those datasets as well.

**DPO Experiments**    Prior to the RFT experiments, we conducted DPO experiments on the ChartQA dataset under the same conditions as described in section 4. However, the improvements were modest, with a 72.3 (+0.5) gain in CoT prediction and a 74.2 (+0.5) gain in direct prediction. In contrast, RFT yielded a significant improvement, with 77.9 (+6.1) on CoT prediction and 74.6 (+0.9) on direct prediction. We hypothesize that for models with relatively weak CoT reasoning capabilities, RFT may be more effective in enhancing model performance, whereas DPO with preference modeling may be less impactful. We leave further analysis for future work.

# E    SFT ABLATION EXPERIMENTS

Table E.1: SFT Ablation Results: For each dataset, '-C' indicates the inclusion of CoT data for training, and '-D' indicates the inclusion of direct prediction data.

| Methods | Prompt | A-OK | ChartQA | DocVQA | InfoVQA | TextVQA | AI2D | SQA | MathVista |
|---|---|---|---|---|---|---|---|---|---|
| LLAVA-NEXT-FORMAT | direct | 85.8 | 70.2 | 75.7 | 37.7 | 68.2 | 71.5 | 75.4 | 39.3 |
| | cot | 84.3 | 71.2 | 67 | 34.9 | 62.2 | 67.4 | 74.4 | 40.3 |
| ChartQA-C+D | direct | 85 | 74.9 | 75.8 | 36.5 | 68.2 | 72.2 | 77.4 | 42.8 |
| | cot | 84.4 | 81.7 | 69 | 32.2 | 63.3 | 68.6 | 74.9 | 41.7 |
| ChartQA-D | direct | 85.2 | 73.1 | 74.6 | 34.1 | 67.1 | 71.5 | 76.4 | 40.3 |
| | cot | 84.3 | 71.8 | 62.4 | 31.8 | 58 | 66.3 | 74 | 35.5 |
| ChartQA-C | direct | 85.1 | 70.8 | 74.5 | 35 | 67.9 | 71.6 | 76.9 | 35.3 |
| | cot | 84.9 | 81.4 | 67.2 | 32.2 | 61.5 | 68.8 | 76.6 | 40.1 |
| A-OK-C+D | direct | 86.2 | 69.2 | 75.4 | 37.7 | 67.3 | 70.7 | 77.5 | 38.8 |
| | cot | 84.6 | 70.2 | 67.3 | 36 | 61.6 | 67.2 | 75.8 | 39.8 |
| A-OK-D | direct | 85.1 | 69 | 75.3 | 38.5 | 66.9 | 72.2 | 76.1 | 39.5 |
| | cot | 84 | 67.7 | 66.5 | 34.8 | 61.1 | 68.4 | 76 | 39.9 |
| A-OK-C | direct | 84.4 | 69.4 | 75.8 | 37.4 | 67.9 | 69.2 | 77.3 | 34.6 |
| | cot | 84.1 | 69.2 | 67.6 | 35.5 | 59.4 | 67.6 | 74.5 | 40.6 |
| DocVQA-C+D | direct | 85.5 | 69.5 | 80.7 | 40.4 | 68.8 | 72 | 77.5 | 41.1 |
| | cot | 83.9 | 70.9 | 80 | 40.2 | 64.1 | 68.2 | 73.4 | 39.3 |
| DocVQA-D | direct | 85.5 | 66.5 | 77 | 39.1 | 68.2 | 70.8 | 76.3 | 41.9 |
| | cot | 83.9 | 66 | 66.4 | 33.7 | 59.9 | 64.8 | 74.5 | 39.3 |
| DocVQA-C | direct | 85.2 | 69.1 | 79.1 | 37.5 | 68.5 | 72 | 76.7 | 33.8 |
| | cot | 84.4 | 71.2 | 78 | 38.5 | 63.5 | 68.5 | 74.1 | 38 |
| InfoVQA-C+D | direct | 85.8 | 63.4 | 77.1 | 47.7 | 67.6 | 72.5 | 78.1 | 43.6 |
| | cot | 85.3 | 65.4 | 72.6 | 47.5 | 62.4 | 69.4 | 74.6 | 37.8 |
| InfoVQA-D | direct | 85.7 | 56.7 | 75 | 45.4 | 67 | 72.5 | 77.5 | 42.8 |
| | cot | 83.7 | 53 | 63.5 | 37.8 | 58.2 | 67 | 75 | 37 |
| InfoVQA-C | direct | 85.2 | 68.3 | 76.5 | 42.5 | 67.8 | 72.5 | 78.2 | 39 |
| | cot | 83.7 | 63.4 | 71.1 | 46.3 | 59.9 | 67.4 | 74.3 | 37.6 |
| TextVQA-C+D | direct | 85.1 | 69.8 | 75.5 | 38.7 | 73 | 71.9 | 76.9 | 42.6 |
| | cot | 84.6 | 68.9 | 70.5 | 36.3 | 70.9 | 67.6 | 76.6 | 36.1 |
| TextVQA-D | direct | 84.9 | 68.6 | 74.5 | 37.6 | 71.8 | 70.8 | 77 | 41.7 |
| | cot | 84.4 | 63.3 | 64.2 | 33.2 | 64.2 | 66.1 | 73.6 | 38.2 |
| TextVQA-C | direct | 84.6 | 69.1 | 74.6 | 36.9 | 71.4 | 71.9 | 77.1 | 36.6 |
| | cot | 84.7 | 68.2 | 69.5 | 36.9 | 70.3 | 67.8 | 75.1 | 37.1 |
| SQA-C+D | direct | 85.7 | 69 | 75 | 38.4 | 67.3 | 72.3 | 90.2 | 38.7 |
| | cot | 83.1 | 71.2 | 66.5 | 35.6 | 58.9 | 66.9 | 90.4 | 40.8 |
| SQA-D | direct | 84.9 | 68.1 | 74.3 | 37 | 66.8 | 72.2 | 89.2 | 41.3 |
| | cot | 83 | 68.4 | 67.5 | 33.8 | 62.1 | 68.7 | 81.9 | 39.8 |
| SQA-C | direct | 84 | 69.3 | 76 | 38.3 | 68.2 | 71.7 | 85 | 39.2 |
| | cot | 82 | 69 | 65.3 | 34.4 | 58.3 | 66.6 | 88.8 | 39.4 |
| AI2D-C+D | direct | 85.2 | 69.6 | 75.8 | 39 | 67.6 | 78 | 78.4 | 40.1 |
| | cot | 83.8 | 70.2 | 68 | 35.9 | 60.7 | 76.3 | 76.6 | 42.1 |
| AI2D-D | direct | 86.3 | 69.2 | 75.1 | 37.3 | 67.2 | 76.8 | 77.6 | 39.7 |
| | cot | 82.7 | 67.6 | 66 | 33.7 | 61.4 | 71.7 | 74.4 | 38.3 |
| AI2D-C | direct | 84.4 | 69.6 | 75.9 | 37.7 | 68.2 | 75 | 76.3 | 39.1 |
| | cot | 83.1 | 70.2 | 65.9 | 35.6 | 59.9 | 75.1 | 74.5 | 39.5 |
| math-C+D | direct | 85.3 | 68.5 | 75.5 | 37.8 | 67.3 | 71.7 | 77.4 | 42.7 |
| | cot | 84.4 | 69.7 | 64.3 | 34.2 | 59.3 | 68.7 | 76.3 | 49 |
| math-C | direct | 85.2 | 68.1 | 75.6 | 38 | 67.4 | 72 | 77.5 | 40.5 |
| | cot | 84.3 | 70.6 | 66.2 | 34.7 | 59.8 | 68.2 | 78.4 | 45.4 |
| math+ChartQA | direct | 85.3 | 70.9 | 75.7 | 36.8 | 67.8 | 71.7 | 78.3 | 41.9 |
| | cot | 84.1 | 81.9 | 67 | 32.6 | 60.7 | 68.3 | 75.5 | 49.7 |
| LLAVA-REASONER-SFT | direct | 85.4 | 76.1 | 82.9 | 50.6 | 73.1 | 79.4 | 90.4 | 44.3 |
| | cot | 86.2 | 83.0 | 81.8 | 51.6 | 71.1 | 78.5 | 92.7 | 50.6 |

In table E.1, we present additional ablation experiments on SFT across each dataset, using three settings: direct only, CoT only, and direct + CoT. Additionally, format-aligned data is incorporated during training to enable the model to follow the specific direct or CoT format during inference.

# F    ADDITIONAL DPO EXPERIMENTS

Table F.1: Truncating response length affects the final performance of DPO. No truncation leads to a decline in performance, while truncating to 90 tokens empirically yields the best results.

| Data/Truncate Len | prompting | 70 | 90 | 110 | No Truncate | SFT baseline |
|---|---|---|---|---|---|---|
| ChartQA | direct | 76.5 | 76.2 | 76.7 | 75.9 | 76.1 |
| | CoT | 83.9 | 84.2 | 81.8 | 80.6 | 83.0 |
| A-OKVQA | direct | 85.2 | 85.2 | 85.3 | 85.1 | 85.4 |
| | CoT | 86.7 | 86.9 | 86.3 | 85.7 | 86.2 |

**Truncating Responses for DPO**    In our initial experiments, we observed that truncating response length impacts the final performance of DPO. As shown in table F.1, no truncation results in a decline in performance, while truncating to 90 tokens empirically produces the best results. Consequently, we applied a 90-token truncation for the DPO experiments.

Table F.2: Comparison of DPO with the RFT method. The upper part of the table presents the SFT baseline and the DPO model, while the lower part shows the ablation results of RFT trained on each of the A-OK, ChartQA, and math training datasets, as well as their combined results.

| Methods | prompting | A-OK | ChartQA | MathVista |
|---|---|---|---|---|
| SFT baseline | direct | 85.4 | 76.1 | 44.3 |
| | CoT | 86.2 | 83.0 | 50.6 |
| LLAVA-REASONER-DPO | direct | 85.4 | 76.4 | 44.2 |
| | CoT | 87.0 | 84.2 | **52.1** |
| A-OKVQA | direct | 85.1 | 72.7 | 37.4 |
| -RFT | CoT | **87.7** | 0.0 | 32.5 |
| A-OKVQA | direct | 85.8 | 74.9 | 41.3 |
| -RFT+Format | CoT | 86.3 | 80.2 | 46.5 |
| ChartQA | direct | 85.4 | 75.0 | 42.6 |
| -RFT | CoT | 86.7 | 83.9 | 52.0 |
| ChartQA | direct | 85.9 | 75.8 | 44.4 |
| -RFT+Format | CoT | 85.5 | 83.4 | 50.6 |
| Math | direct | 85.3 | 76.0 | 32.4 |
| -RFT | CoT | 86.7 | 67.3 | 50.9 |
| Math | direct | 85.5 | 76.0 | 39.6 |
| -RFT+Format | CoT | 85.5 | 82.0 | 50.0 |
| Combined | direct | 85.3 | 75.4 | 37.8 |
| -RFT | CoT | 85.4 | **84.4** | 49.0 |
| Combined | direct | 85.0 | 75.5 | 43.0 |
| -RFT+Format | CoT | 86.6 | 83.1 | 47.1 |

**DPO vs. RFT**    Following appendix D, we examine the impact of RFT and compare it to the DPO method.

In table F.2, for A-OKVQA, we observe that training with A-OKVQA RFT alone yields the best result for A-OKVQA; however, the model's ability to generate short answers is entirely lost. When format-aligned data is added, there is a trade-off between performance on A-OKVQA and other datasets.

When the datasets are combined for training, we see improvements only on ChartQA, while performance on A-OKVQA and MathVista declines. This indicates that balancing RFT across datasets is challenging, especially when the SFT model already performs relatively well on basic tasks. In contrast, the DPO model demonstrates consistent gains across datasets, showing better generalization.

