# OpenReview forum: "Improve Vision Language Model Chain-of-thought Reasoning"
_ICLR.cc/2025/Conference — ICLR 2025 Conference Withdrawn Submission_

### Official Review · Reviewer_E36p · 2024-10-25

**Soundness:** 2
**Presentation:** 3
**Contribution:** 2
**Rating:** 3
**Confidence:** 3

**Summary:**

This paper aims to equip vision-language models with the capability for chain-of-thought (CoT) reasoning. To achieve this, this paper first prompts GPT-4o to generate rationales as CoT training data for supervised fine-tuning of VLM. Then, Direct Preference Optimization (DPO) is used to optimize the VLM, where the positive data consists of generated rationales that lead to correct answer, while the negative data consists of generated rationales that lead to incorrect answer. The two-staged training method respectively improves the accuracy across several ImageQA benchmarks in both direct prompting and CoT prompting setting.

**Strengths:**

- The paper is well-written and thus easy to understanding.
- The motivation to equip VLMs with CoT capability is interesting.
- The two-stage method demonstrates improved accuracy across several ImageQA benchmark.

**Weaknesses:**

[Setting]
1. Lack of discussion on the reasons why the VLM lacks CoT reasoning capabilities, even when its LLM backbone possesses CoT capabilities. If the LLM backbone has already equipped with CoT capability, why does the VLM not retain this capability? On the other hand, if the LLM backbone itself lacks CoT capability, it raises the question of why improvements were not made directly to the CoT capabilities of the LLM.
2. Does llama3-8b have CoT capabilities (I ask this question because your base VLM is LLaMA3-LLaVA-NeXT-8B)? If not, why was the approach not to directly improve CoT capabilities within llama3-8b? If yes, why does llama3-8b seem to lack CoT capability here, given that its pre-training text-only corpus is 15 TB [1], which significantly larger than the image-caption pairs used for VLM pre-training?

[Method]
1. In the DPO optimization stage, how to ensure that the correctness of the predicted answer is genuinely due to the rationale provided by the model, rather than the VLM simply gets the correct answer from its parameteric knowledge and directly ignores the incorrect rationale?
2. What proportion of the VLM's correct answers are generated with an incorrect rationale?
3. The two-stage training design, in which knowledge distillation with a larger model is followed by reinforcement learning optimization, is similar to approaches in previous studies [2, 3]. It would be beneficial to clearly articulate how this training design differs from those outlined in prior works.


[1] The Llama 3 Herd of Models (arXiv 2408)

[2] Rainier: Reinforced Knowledge Introspector for Commonsense Question Answering (EMNLP 2022)

[3] RAPPER: Reinforced Rationale-Prompted Paradigm for Natural Language Explanation in Visual Question Answering (ICLR 2024)

**Questions:**

Please answer my questions in “weaknesses” section.

---

> ### Author Response · Authors · 2024-11-25
>
> Thank you for taking the time to review our paper! We are delighted that you found value in the key contributions of our work, particularly the motivation of research and effectiveness of methods.
>
> We believe your concerns can be addressed with the following clarifications. Please find our detailed responses below. We would be happy to engage further during the discussion period if any points remain unclear.
>
> ---
>
> # Setting
>
> ***Why doesn't VLM have CoT?***
>
> The LLaMA3-LLaVA-Next-8B model is initialized from the LLaMA3-8B-Instruct checkpoint, which possesses CoT reasoning capabilities. However, its VLM counterpart tends to underperform in CoT tasks. Qualitative failure cases for LLaVA-LLaMA3-8B are provided in Appendix C, while Table 1 presents quantitative evidence showing poor CoT performance without specific CoT training.
>
> We hypothesize two primary reasons for this gap:
>
> 1.	Task-Specific Generalization: CoT evaluation in the VL domain involves tasks like enumerating items in a figure or diagnosing diseases from an ECG. These capabilities are not
> adequately covered by text-only pretraining, making generalization from text to vision challenging.
>
> 2.	Attention Dynamics: In the VLM setup, LLaVA introduces about 2000 visual tokens as additional input to the LLM. This may alter the attention mechanism and affect feature extraction for reasoning tasks.
>
> Our study takes a data-driven approach to improve VLM CoT reasoning in an end-to-end manner. However, decoupling reasoning between vision and LLM components is beyond the scope of this work.
>
>
> # Method
>
> ***How to Avoid VLM Ignoring Rationale?***
>
> Our experiments reveal that truncating reasoning to 90 tokens (for an average output length of 120 tokens) is essential for improvement. Detailed analysis is provided in Appendix F. In this setup, the final answers are typically excluded from DPO preference pairs. To ensure consistency, we removed the final answer section (e.g., “### Answer: …”) after the rationale during training. This truncation trick had no observable effect on performance.
>
> ***Proportion of Correct Answers with Incorrect Rationales***
>
> We do not compute this statistic due to the lack of efficient and reliable methods for evaluating rationale correctness. Instead, we propose outcome-based rewards inspired by [1, 2], which provide a scalable, albeit imperfect, way to assess rationale quality. While the resulting pairwise data can be noisy, our benchmarks show moderate yet consistent performance gains.
>
> [1] Zhiqing Sun, Longhui Yu, Yikang Shen, Weiyang Liu, Yiming Yang, Sean Welleck, and Chuang Gan. Easy-to-hard generalization: Scalable alignment beyond human supervision
>
> [2] Arian Hosseini, Xingdi Yuan, Nikolay Malkin, Aaron Courville, Alessandro Sordoni, and Rishabh Agarwal. V-star: Training verifiers for self-taught reasoners
>
>
> ***Differences from Previous Works***
>
> Our work differs significantly from prior approaches like Rainier and RAPPER, both of which adopt a two-stage framework but focus on different tasks and goals:
> 1. Rainier addresses commonsense reasoning in the text domain.
> 2. RAPPER is limited to explanation of commonsense knowledge in the VQA domain.
>
> In contrast, our work targets CoT in the VL domain, encompassing a broader range of contexts, such as world knowledge, chart interpretation, and information localization in documents. We define a more extensive range of tasks and demonstrate that leveraging a short-annotation dataset (combined with distillation and outcome-based rewards) improves CoT performance across nearly all tasks.
>
> Additionally, our implementation differs in several technical details:
>
> 1.	Reward Design: Prior works focus on LM probability, while we adopt outcome-based rewards.
> 2.	Algorithm Choice: We use DPO instead of PPO.
> 3.	Evaluation Settings: Our evaluation setup accounts for the challenges of VL reasoning with relatively new benchmarks.

---

> > ### Comment · Reviewer_E36p · 2024-11-27
> > **Official Comments by  Reviewer E36p**
> >
> > **Why doesn't VLM have CoT?**
> > >The LLaMA3-LLaVA-Next-8B model is initialized from the LLaMA3-8B-Instruct checkpoint, which possesses CoT reasoning capabilities. However, its VLM counterpart tends to underperform in CoT tasks. Qualitative failure cases for LLaVA-LLaMA3-8B are provided in Appendix C, while Table 1 presents quantitative evidence showing poor CoT performance without specific CoT training.
> >
> > Table1 is not quantitative results.
> >
> > >We hypothesize two primary reasons for this gap: (1) Task-Specific Generalization: CoT evaluation in the VL domain involves tasks like enumerating items in a figure or diagnosing diseases from an ECG. These capabilities are not adequately covered by text-only pretraining, making generalization from text to vision challenging.
> >
> > This claim does not convince me due to the lack of concrete examples. I personally think that enumerating items in a figure or diagnosing diseases from an ECG can also be a more complex image captioning tasks. (Please correct me if I am wrong)
> >
> > > (2) Attention Dynamics: In the VLM setup, LLaVA introduces about 2000 visual tokens as additional input to the LLM. This may alter the attention mechanism and affect feature extraction for reasoning tasks.
> >
> > This also lacks supportive evidence. The authors are required to show some experiment results to support this claim such as attention visualization.
> >
> > **How to Avoid VLM Ignoring Rationale?**
> > >Our experiments reveal that truncating reasoning to 90 tokens (for an average output length of 120 tokens) is essential for improvement. Detailed analysis is provided in Appendix F. In this setup, the final answers are typically excluded from DPO preference pairs. To ensure consistency, we removed the final answer section (e.g., “### Answer: …”) after the rationale during training. This truncation trick had no observable effect on performance.
> >
> > Explaining the truncation trick does not address my concern (i.e., how to Avoid VLM Ignoring Rationale?).
> >
> > **Proportion of Correct Answers with Incorrect Rationales**
> > >We do not compute this statistic due to the lack of efficient and reliable methods for evaluating rationale correctness. Instead, we propose outcome-based rewards inspired by [1, 2], which provide a scalable, albeit imperfect, way to assess rationale quality. While the resulting pairwise data can be noisy, our benchmarks show moderate yet consistent performance gains.
> >
> > Explaining the outcome-based rewards again doesn't address my concern (i.e., the proportion of correct answers with incorrect rationales). On the other hand, assessing the rationale correctness by using arbitrary VLM (e.g. LLaVA [1]) is the simplest solution.
> >
> > Overall, I believe the responses provided do not fully resolve some of the major concerns I raised. While the paper proposes novel idea of equipping VLMs with CoT capability, it seems that more evidence and concrete analysis are required to make it ready for publication. Thus, I have decided to decrease my rating to 3 accordingly.

---

### Official Review · Reviewer_TsHB · 2024-11-01

**Soundness:** 3
**Presentation:** 3
**Contribution:** 2
**Rating:** 5
**Confidence:** 4

**Summary:**

This work focuses on enhancing the CoT capabilities of VLMs to serve complex vision-language reasoning tasks. The proposed approach includes generating synthetic data, SFT for basic CoT capability, and DPO for further calibration of complex reasoning ability stages.

The method is overall reasonable. With results on 9 datasets, the authors demonstrate the effectiveness of the proposed method. The experiments are clear and the analysis is comprehensive. Overall, this work positively contributes to the applications that involve VLM reasoning.

**Strengths:**

1. The overall design makes sense. As demonstrated by the benchmark results, this could be a possible choice for implementing vision-language applications.
2. The release of synthetic data generated by GPT-4o contributes to the VLM finetuning.
3. The failure analysis in this paper is insightful and potentially helpful for other VLM research.

**Weaknesses:**

The novelty is unclear.
- From the idea level, tuning for CoT ability is being actively explored in LLM-centric research (e.g., [1], [2], [3])
- For the overall design, the proposed method has recently been widely used. Including,
   - Step 1 directly leverages the commonly seen approach, i.e., knowledge distillation from a larger teacher model (GPT-4). (e.g., [4])
   - Two-stage tuning (SFT-RL) has been shown to be effective in improving reasoning ability in many LLM works, either through
       - Pertaining (e.g., Llama-3)
       - SFT on synthetic CoT data, then use RL to refine the LLM's reasoning abilities (e.g., [5])

Besides, one minor problem is missing related work in language-based reasoning.
Specifically, "How does LLM research tackle the author-targeted challenges in pure language tasks?" This background discussion would be beneficial because this paper particularly studies the Vision-Language Model, which utilizes a vision encoder to project visual information onto language space; thus, it may share certain characteristics with its underlying LLM.

Reference:

[1] Beyond Human Data: Scaling Self-Training for Problem-Solving with Language Models. https://arxiv.org/pdf/2312.06585

[2] LogiCoT: Logical Chain-of-Thought Instruction Tuning. https://arxiv.org/pdf/2305.12147

[3] RL on Incorrect Synthetic Data Scales the Efficiency of LLM Math Reasoning by Eight-Fold: https://arxiv.org/pdf/2406.14532

[4] LLaVA-Plus: Learning to Use Tools for Creating Multimodal Agents. https://arxiv.org/pdf/2311.05437

[5] Tackling Vision Language Tasks Through Learning Inner Monologues. https://arxiv.org/pdf/2308.09970

**Questions:**

Given the concerns raised in the Weakness section, more discussion on each point summarized in lines 88-92 would help to showcase the novelty and contributions of the work.

To be more clear, lines 88-92 claim the contribution is to (A) provide a GPT-generated dataset, (B) verify that this dataset can be used for SFT, and (C) use DPO after SFT through an approach similar to [3] is meaningful. This makes me feel that this work is more of a resource paper. I would like to learn more about the technical contributions.

Reference:

[3] RL on Incorrect Synthetic Data Scales the Efficiency of LLM Math Reasoning by Eight-Fold: https://arxiv.org/pdf/2406.14532

---

> ### Author Response · Authors · 2024-11-25
>
> Thank you for taking the time to review our paper! We are delighted that you found value in the key contributions of our work, particularly the validity of design, release of data and our experiments and ablations on SFT and DPO methods.
>
> We believe your concerns can be addressed with the following clarifications. Please find our detailed responses below. We would be happy to engage further during the discussion period if any points remain unclear.
>
> ---
>
>
> # 1. Research question and novelty (Question 1)
> The predominance of short-answer training datasets leaves several critical research questions about vision-language (VL) reasoning unanswered. Our paper addresses two key questions:
>
> ***A. Can direct supervision implicitly improve reasoning capabilities? How does it compare to training on CoT data?***
>
> We find that training on short-answer datasets yields limited improvements in Chain-of-Thought (CoT) reasoning, especially for tasks requiring complex operations such as calculations. This limitation is evident in datasets like ChartQA (OCR + calculation) and MathVista (visual math). Fine-tuning (SFT) on our CoT dataset significantly enhances reasoning capabilities across various benchmarks, demonstrating strong generalization even on challenging datasets like MMMU.
>
> ***B. How can reasoning correctness be effectively assessed?***
>
> The development of effective and efficient methods for evaluating reasoning in VL tasks has been underexplored. Drawing inspiration from outcome-based rewards in domains such as code generation and mathematical reasoning (e.g., [1, 2]), we extend this approach to a wider range of VL reasoning tasks. Our findings show that:
>
> 1.	Outcome-based rewards effectively identify and distill higher-quality training data for VL reasoning.
>
> 2.	Integrating outcome-based rewards for DPO achieves moderate yet consistent performance improvements, as further discussed in the following section.
>
>
> [1] Zhiqing Sun, Longhui Yu, Yikang Shen, Weiyang Liu, Yiming Yang, Sean Welleck, and Chuang Gan. Easy-to-hard generalization: Scalable alignment beyond human supervision
>
> [2] Arian Hosseini, Xingdi Yuan, Nikolay Malkin, Aaron Courville, Alessandro Sordoni, and Rishabh Agarwal. V-star: Training verifiers for self-taught reasoners
>
>
> # Difference between LLM CoT
>
> CoT reasoning in LLMs primarily focuses on domains like math and code, as highlighted in [1, 2]. In the VL domain, however, CoT reasoning encompasses a broader range of contexts, including world knowledge, chart interpretation, information localization in documents, and more. Our work defines a more extensive range of tasks than previous studies (e.g., [3, 4]) and demonstrates that leveraging a short-annotation dataset (with distillation and outcome reward) improves CoT performance across nearly all tasks.
>
> We will add discussion on LLM reasoning such as [1,2] and the ones you included in the review in related work.
>
>
> [3] Bozhi Luan, Hao Feng, Hong Chen, Yonghui Wang, Wengang Zhou, and Houqiang Li. Textcot: Zoom in for enhanced multimodal text-rich image understanding
>
> [4]Hao Shao, Shengju Qian, Han Xiao, Guanglu Song, Zhuofan Zong, Letian Wang, Yu Liu, and Hongsheng Li. Visual cot: Unleashing chain-of-thought reasoning in multi-modal language models.

---

### Official Review · Reviewer_yp44 · 2024-11-02

**Soundness:** 2
**Presentation:** 1
**Contribution:** 2
**Rating:** 3
**Confidence:** 4

**Summary:**

The paper focuses on enhancing chain-of-thought (CoT) reasoning in vision-language models (VLMs) to improve their interpretability and performance on complex tasks. The authors identify that current training methods often rely on datasets with short annotations and limited rationales, which restrict the models' ability to generalize to tasks requiring detailed reasoning.

To address these limitations, the authors propose a two-fold approach:

1. Data Distillation: They distill rationales from the GPT-4o model to enrich the training data, creating a comprehensive CoT dataset with 193,000 examples across various visual question answering (VQA) tasks.

2. Reinforcement Learning (RL): They apply reinforcement learning, specifically the Direct Preference Optimization (DPO) algorithm, to refine the reasoning abilities of the models by constructing positive (correct) and negative (incorrect) reasoning pairs based on model-generated outputs.

**Strengths:**

A clear two-stage approach is proposed to improve the chain-of-thought reasoning and the final performance of vision-language models. The overall research goal and method are easy to understand.

**Weaknesses:**

1. The novelty in the proposed two-stage approach is quite limited, and similar strategies have been widely adopted in previous work. For the first stage that distills CoT reasoning chain from the teacher models GPT-4o, this kind of method has been implemented too many of times in different domains, different modality, and different purposes. For the second stage that prompt VLM to first generate the CoT solution and then check the correctness of the final answers by comparing with the ground-truth ones, it's first proposed in STaR [1] and has been widely extended in many following work. Although the authors leverage the DPO for further optimization, it is far from a shinning point to make the whole method novel.
2. I doubt the scalability of the proposed method. For the first stage, it basically relies on a teacher model to provide supervision. For the second stage, it needs ground-truth annotation from existing supervised datasets, which makes the method less suitable for more open-domain and complex problems.
3. The results in Table 2 and Table 6 are not impressive. The improvement is little and hard to tell whether it's because of distilling more GPT data than previous work.


[1] STaR: Bootstrapping Reasoning With Reasoning. Eric Zelikman et al

**Questions:**

N/A

---

> ### Author Response · Authors · 2024-11-25
>
> Thank you for taking the time to review our paper! We are delighted that you found value in the key contributions of our work, particularly the distillation on the CoT data, plus the usage of data on RL ( DPO) methods.
>
> We believe your concerns can be addressed with the following clarifications. Please find our detailed responses below. We would be happy to engage further during the discussion period if any points remain unclear.
>
> ---
>
> # 1. Research question and contribution (weakness 1)
> The predominance of short-answer training datasets leaves several critical research questions about vision-language (VL) reasoning unanswered. Our paper addresses two key questions:
>
> ***A. Can direct supervision implicitly improve reasoning capabilities? How does it compare to training on CoT data?***
>
> We find that training on short-answer datasets yields limited improvements in Chain-of-Thought (CoT) reasoning, especially for tasks requiring complex operations such as calculations. This limitation is evident in datasets like ChartQA (OCR + calculation) and MathVista (visual math). Fine-tuning (SFT) on our CoT dataset significantly enhances reasoning capabilities across various benchmarks, demonstrating strong generalization even on challenging datasets like MMMU.
>
> ***B. How can reasoning correctness be effectively assessed?***
>
> The development of effective and efficient methods for evaluating reasoning in VL tasks has been underexplored. Drawing inspiration from outcome-based rewards in domains such as code generation and mathematical reasoning (e.g., [1, 2], and STaR), we extend this approach to a wider range of VL reasoning tasks. Our findings show that:
>
> 1.	Outcome-based rewards effectively identify and distill higher-quality training data for VL reasoning.
>
> 2.	Integrating outcome-based rewards for DPO achieves moderate yet consistent performance improvements, as further discussed in the following section.
>
>
>
>
> [1] Zhiqing Sun, Longhui Yu, Yikang Shen, Weiyang Liu, Yiming Yang, Sean Welleck, and Chuang Gan. Easy-to-hard generalization: Scalable alignment beyond human supervision
>
> [2] Arian Hosseini, Xingdi Yuan, Nikolay Malkin, Aaron Courville, Alessandro Sordoni, and Rishabh Agarwal. V-star: Training verifiers for self-taught reasoners
>
> # 2. Contribution on Data Resources (weakness 1)
>
> Existing Chain-of-Thought (CoT) distillation methods predominantly focus on the math and text domains. In contrast, our work targets the VL domain, where CoT reasoning involves the integration of both visual and textual information. To facilitate comprehensive experimentation on CoT learning, we curate a diverse collection of VL reasoning datasets, encompassing nine datasets and a total of 193k examples across a wide range of domains.
>
> # 3.Does distillation count as a technical contribution? (weakness 1,3)
>
> We require CoT data to address the research questions outlined above, and therefore, we do not position the use of GPT-4o as a primary contribution. GPT-4o is simply a practical approach to generating large-scale data, addressing the existing gap in datasets suitable for VL reasoning research. The data we curate serves as a foundational resource or baseline for future research in CoT-based reasoning within the vision-language domain.
>
>
> # 4. Effective of data/methods (weakness 3)
>
> ***SFT:***
>
> Table 2 demonstrates the effectiveness of supervised fine-tuning (SFT). Model 4 (our SFT model) achieves a +8.8% improvement in CoT performance compared to the LLaVA-Next-8b baseline (Model 1). Furthermore, in Table 5, our SFT model achieves a +4.3% performance gain over Cambrian, which uses 7M SFT training samples, despite leveraging fewer than 400k samples. These results underscore the efficiency and effectiveness of our curated CoT dataset in improving model performance.
>
> ***DPO:***
>
> Table 4 shows that Direct Preference Optimization (DPO) applied to samples from three datasets delivers a +0.9% improvement over SFT. While the performance gains are moderate, they are consistent across benchmark datasets. To ensure a fair comparison, we evaluate against a model trained with RLAIF data. Our results indicate that the DPO-trained model not only excels at greedy decoding but also improves ranking capabilities, as illustrated in Figures 6 and 7.

---

> ### Author Response · Authors · 2024-11-25
>
> ***(continue)***
> # 5.Scalability of the Method (Weakness 2)
>
> Relying on short-answer questions does not limit scalability. On the contrary, short-answer data is significantly easier to acquire than long-answer formats, and its correctness can be more efficiently verified using outcome-based rewards. Additionally, data augmentation techniques like those used in STaR can be applied to short-answer datasets to generate large quantities of preference pairs. For comparison, we use only three datasets to produce 66k DPO training samples. Scaling up to all nine datasets could yield over 200k preference pairs, further enhancing the scalability of our approach. In comparison, LLaVA-RLHF has 10k human labeled preference pairs, and RLAIF has about 80k pairs.

---

> > ### Comment · Reviewer_yp44 · 2024-11-25
> >
> > Thanks for the reply. I am curious about the contributions of the two research questions you are considering.
> >
> > 1. Can direct supervision implicitly improve reasoning capabilities? How does it compare to training on CoT data?
> >
> > Please correct me if I have any misunderstanding. I personally think this question is quite obvious, given a lot of previous work like STaR-series, ultraInteract. Why do you think this question is interesting to investigate, and what is the remaining part of this question that we still need to investigate?
> >
> > 2. How can reasoning correctness be effectively assessed?
> >
> > You say you draw inspiration from code/math, and "extend this approach to a wider range of VL reasoning tasks".  Do you think trying a well-established method in a different domain (NLP -> VL) is a significant contribution worth highlighting?

---

> > > ### Author Response · Authors · 2024-11-25
> > >
> > > Thanks for your feedback! We would like to address the questions regarding our contributions.
> > >
> > > ***Discussion on VLM CoT vs. LLM CoT***
> > >
> > > In LLMs, CoT reasoning primarily focuses on tasks like mathematics, programming, and logic. In contrast, CoT reasoning for Vision-Language Models (VLMs) encompasses a broader range of contexts, such as leveraging world knowledge, interpreting charts, localizing information within documents, and more. Our work defines such a diverse set of tasks and demonstrates that leveraging datasets with short annotations, combined with distillation and reward-based fine-tuning, enhances CoT performance across nearly all tasks.
> > >
> > > ***Comparing VLM CoT with Direct Training***
> > >
> > > We build upon state-of-the-art open-source VLMs such as Cambrian [1] and LLaVA-OneVision [2], which primarily utilize training datasets designed for short-answer QA. Long-form answers in existing datasets, like those from ShareGPT4V, are often caption-based rather than reasoning-focused.
> > > This raises an important question: can such training data effectively enhance VL CoT capabilities, such as object counting, information localization, or reasoning with world knowledge based on visual input? Our objective is to demonstrate improved reasoning capabilities with respect to the current training paradigm. The datasets and experimental results presented in our paper provide a foundation for further exploration in this direction.
> > >
> > > ***Transferring Methods to Evaluate Reasoning Correctness***
> > >
> > > We do not claim the transfer of an existing method to a new domain as a core contribution. Instead, our contribution lies in empirically validating the utility of such methods and benchmarking them against other state-of-the-art techniques, such as RLAIF. The strength of outcome reward lies in its efficiency and scalability, making it well-suited for this context. Our work verifies its applicability in VLM post-training, covering a wide range of visual reasoning tasks.
> > >
> > > [1] Tong, Shengbang, et al. "Cambrian-1: A fully open, vision-centric exploration of multimodal llms."
> > > [2] Li, Bo, et al. "Llava-onevision: Easy visual task transfer."

---

### Official Review · Reviewer_THp5 · 2024-11-03

**Soundness:** 2
**Presentation:** 3
**Contribution:** 3
**Rating:** 6
**Confidence:** 3

**Summary:**

This paper presents a method for training vision language models (VLMs) to output a chain of thought (CoT) before answering a question. The approach involves creating a dataset with CoTs generated by GPT-4o, finetuning an open-weight VLM on these chains of thought (and the associated answers), and further training the VLM using DPO, where the preferences come from the GPT-generated annotations. The first experiments evaluate the model trained by supervised finetuning on the CoT dataset to several baselines--the model trained w/o CoT, ablated versions where certain parts of the data are removed, and models from prior work. The later experiments evaluate the impact of the DPO procedure and compare it to prior work on applying DPO to VLMs.

**Strengths:**

- The dataset of CoTs is likely to be useful in future work
- The results show that finetuning on CoTs is helpful in several tasks
- The results show marginal improvements from the DPO process
- There are several ablations on dataset composition and methods of training/evaluating the model, which are helpful

**Weaknesses:**

- The improvements from DPO (in Tables 6, 7 and Figure 7) are quite small. (Are they statistically significant?)
- The related work section does not discuss the mentioned work in enough detail (and could probably benefit from inclusion of more related work). For instance, what are the differences between the method in this paper and methods in the prior work that finetunes VLMs on CoTs? The statement that "Shao et al. (2024) trains VLMs for chain-of-thought (CoT) reasoning in object localization tasks" does not seem entirely accurate; their CoTs may involve regions in the image, but they apply their models to VQA tasks.

**Questions:**

- What does format (1) "format alignment only" mean? How does it differ from (2) "direct responses only"? Does (1) involve finetuning on the same data as the other models?
- Similar to the last question, in Tables 3 and 4, which datasets are used to train the "format (1)" model?
- This method is bottlenecked by the performance of the teacher model (in this case, GPT-4o). In this sense, it is a distillation method rather than a method that could be used to advance the performance of any model (e.g. a frontier model).

---

> ### Author Response · Authors · 2024-11-21
>
> Thank you for taking the time to review our paper! We are delighted that you found value in the key contributions of our work, particularly the utility of CoT data as data resource, our experiments and ablations on SFT and DPO methods.
>
> We believe your concerns can be addressed with the following clarifications. Please find our detailed responses below. We would be happy to engage further during the discussion period if any points remain unclear.
>
> ---
>
> ### Revision
> We uploaded a revision to correct related work on Visual-CoT(Shao et al), and add additional prompts, examples and experiments in the appendix. We include clarification on the questions and will add more discussion on DPO and other experiment highlights and contribution.
>
> ---
>
> ### Clarifications on the Experimental Settings
>
> **1. What does format (1), "format alignment only," mean?**
>
> In Figure 5 (https://ibb.co/wMrSHVH), we describe the data mixture used to train the models:
>
> - *Format alignment only* (model 1): This model is trained using 450 CoT data points and 450 direct prediction data points, along with a small random subset of data from LLaVA-Next-8B pretraining.
> - This model achieves performance comparable to LLaVA-Next-8B while supporting both direct prediction and CoT formats. We include examples in the appendix (or view at: https://ibb.co/F6CxVQR)
>
> **How does this differ from (2) "direct responses only"? Does (1) involve fine-tuning on the same data as the other models?**
>
> - *Direct responses only* (model 2): This model includes all the data used in model (1) (450 CoT, 450 direct prediction data points, small random subset of the pretraining data.), along with the rest of direct prediction data.
> - In essence, model (2) is an extension of model (1) with the addition of all direct prediction data.
>
> **In Tables 3 and 4, which datasets are used to train the "format (1)" model?**
>
> The dataset for model (1) is consistent across the paper and is based on the "format alignment only" configuration described above.
>
> **Is this method bottlenecked by the performance of the teacher model?**
>
> We include additional experiments in Appendix D (also available at: https://ibb.co/9p8M74z).
> The model training uses only 450 CoT format data from GPT-4o for format alignment purposes. We perform reject sampling and show significant improvements on CoT performance, though it does not achieve the same effectiveness as training on GPT-4 distilled data. Additionally, we discuss DPO under this setting but observe smaller gains compared to reject sampling. We hypothesize that weaker models may benefit more from the SFT stage prior to applying RL.
>
> Another note: We aim to study specific model behaviors in relation to data and training algorithms. For instance, our experiments verify that training on direct data (predicting the number of items directly) does not implicitly help models acquire CoT reasoning skills (e.g., ability to correctly enumerate items). In contrast, incorporating high-quality CoT data proves significantly beneficial. Such high-quality CoT data can be obtained through methods like GPT-4o distillation, reject sampling (as discussed above), or human-curated data, where feasible.

---

> > ### Comment · Reviewer_THp5 · 2024-11-22
> > **Reply**
> >
> > Overall, I'm inclined to leave my score the same or to decrease it to a 5. For now, I am leaving it as is; my thoughts are below.
> >
> > With respect to related work, it seems that the contribution of this work is to collect and finetune on CoTs for *a wider variety* of visual tasks. Previous papers did so for math/chart questions, while this paper does so also for InfoVQA/DocVQA/TextVQA/SQA. That being said, the improvements from CoT on these non-math and chart datasets seem inconsistent.
> >
> > Plus, as I commented in my original review, this approach is bottlenecked by the quality of the teacher model. In your reply, you say "The model training uses only 450 CoT format data from GPT-4o for format alignment purposes."
> >
> > However, the paper says (lines 144 to 149):
> > "To mitigate the limited availability of high-quality CoT data, we leverage VQA datasets with short annotations and augment them with rationales generated by the GPT-4o model. We collect 193k visual CoT instances to create the SHAREGPT-4O-REASONING dataset, which we plan to release for public use."
> > You also indicate in Figure 5 that these 193k CoT instances are used in settings 3 and 4 of your experiments (which are the best performing.
> >
> > Thus, it does seem like the 193k instances are important for achieving the best results you report, and so it follows that this approach is bottlenecked by the quality of the teacher model.

---

> > > ### Author Response · Authors · 2024-11-25
> > >
> > > Thank you for your feedback. We would like to clarify the experimental settings and share our thoughts on the teacher model.
> > >
> > > ***Clarification on Experiments***
> > >
> > > In the earlier discussion regarding models trained on 450 CoT samples, we referred to the experiment added in Appendix D of the revised manuscript. This experiment demonstrates reject sampling using only CoT-formatted data. Comparing the results with those in the main paper, for example on the ChartQA dataset, we observe the following:
> > >
> > > 	•	Baseline (Direct + format): 71.8% accuracy
> > > 	•	Baseline + Reject sampling: 77.9% accuracy
> > > 	•	SFT on all CoT data: 83.0% accuracy
> > > 	•	SFT + DPO: 84.2% accuracy
> > >
> > > These experiments aim to explore how to use outcome-based rewards to enhance CoT performance when only minimal CoT data is available.
> > >
> > > ***Is the Approach Bottlenecked by the Teacher Model?***
> > >
> > > The experiments in Appendix D indicate that this method can improve CoT performance even without access to a strong teacher model (e.g., using 450 human-labeled samples, which is manageable). This highlights the effectiveness of outcome-based rewards in VLMs. However, the results also suggest that providing additional CoT data would lead to further significant improvements.

---

### Note · Authors · 2024-12-16

**Comment:**

Thanks for the reviewers and meta-reviewers effort! This paper is resubmitted.

**Withdrawal Confirmation:**

I have read and agree with the venue's withdrawal policy on behalf of myself and my co-authors.